# Dihydroartemisinin imposes positive and negative regulation on Treg and plasma cells via direct interaction and activation of c-Fos

Qilong Li[1,2,3], Ning Jiang[1,2,3], Yiwei Zhang[1,2], Yize Liu [1,2], Ziwei Su[1,2], Quan Yuan[1,2], Xiaoyu Sang[1,2], Ran Chen[1,2], Ying Feng[1,2] & Qijun Chen [1,2✉]

Dihydroartemisinin (DHA), a potent antimalarial drug, also exhibits distinct property in modulation on $T_{reg}$ and B cells, which has been recognized for decades, but the underlying mechanisms remain understood. Herein we revealed that DHA could promote $T_{reg}$ proliferation, meanwhile, suppress B cell expansion in germinal centers, and consequently decrease the number of circulating plasma cells and the content of serum immunoglobulins. Further, DHA-activated $T_{reg}$ significantly mitigated lipopolysaccharide-induced and malaria-associated inflammation. All these scenarios were attributed to the upregulation of c-Fos expression by DHA and enhancement of its interaction with target genes in both $T_{reg}$ and circulating plasma cells with bilateral cell fates. In $T_{reg}$, the c-Fos-DHA complex upregulated cell proliferation-associated genes and promoted cell expansion; whereas in plasma cells, it upregulated the apoptosis-related genes resulting in decreased circulating plasma cells. Thus, the bilateral immunoregulatory mechanism of DHA was elucidated and its application in the treatment of autoimmune diseases is further justified.

[1] Key Laboratory of Livestock Infectious Diseases, Ministry of Education, Key Laboratory of Zoonosis, College of Animal Science and Veterinary Medicine, Shenyang Agricultural University, 120 Dongling Road, Shenyang 110866, China. [2] Research Unit for Pathogenic Mechanisms of Zoonotic Parasites, Chinese Academy of Medical Sciences, 120 Dongling Road, Shenyang 110866, China. [3] These authors contributed equally: Qilong Li, Ning Jiang. ✉email: qijunchen759@syau.edu.cn

Artemisinin (ART) and its derivative dihydroartemisinin (DHA) have demonstrated sensational therapeutic efficacy in cancers, lupus nephritis, and lupus erythematosus in humans and murine models[1–3]. In detail, the antitumor effect of DHA was recently suggested to be due to its promotion on human γδ T cell proliferation[4]. A previous report study suggested that DHA promoted $T_{reg}$ differentiation and increased the proportion of effector $T_{reg}$ in the spleen, which is likely the reason for the progression inhibition of lupus nephritis after administration[5,6]. Additionally, DHA was also found to markedly suppress humoral responses in both mice and humans[6–9]. Thus, the bilateral property of DHA in immunoregulation on splenic $T_{reg}$ and plasma cells has been postulated, while the underlying mechanism remains elucidated.

$T_{reg}$ cells are indispensable for their prominent role in maintaining immune tolerance and homeostasis by limiting the autoreactive T cells and reducing inflammatory responses[10,11]. Importantly, they can exert positive or negative regulation by either enhancing or weakening immune responses via down-regulating various biological functions[12]. $T_{reg}$ cells express specific surface markers, including CD25[13], CD73, and cytotoxic T lymphocyte antigen-4 (CTLA-4, CD152)[14], which suppress most of the immune effector cells, including CD4$^+$ T cells, CD8$^+$ cytotoxic T cells, NK cells, NKT cells, macrophages, dendritic cells, neutrophils, B cells[15,16]. $T_{reg}$ cells can also function to inhibit the germinal center reaction and antibody generation[17]. However, $T_{reg}$-mediated immune repression, by the release of soluble mediators and $T_{reg}$-associated cytokines such as IL-10 and 35, TGF-β, is obviously a *rapier*; the operation of maintaining self-tolerance is in most time not discriminative, they also suppress host immunity to anomalies such as tumorous cells or tissues[17,18]. In certain circumstances, $T_{reg}$ activity may need to be restricted for the engagement of effector cells or control malignant cell expansion. Thus, chemicals that can beneficially modulate $T_{reg}$-mediated immune regulation are much desired for the treatment of a variety of diseases.

Strategies for $T_{reg}$-based therapy for autoimmune diseases are to enhance the $T_{reg}$ expansion to achieve maximal suppressive effect or to limit their locate presence, such as in cancer and other autoimmune disorders, including forkhead box protein 3 (Foxp3) disruption-based $T_{reg}$ removal and CTLA-4 antibody-based effector $T_{reg}$ blockade[19–21]. The Foxp3, a master transcription factor and also a cell signature, is required in most $T_{reg}$ lineages for their stability and functionality[22]. Foxp3 requires interaction with a myriad protein factors like AML/Runx1 (acute myeloid leukemia 1/Runt-related transcription factor 1), NFAT (nuclear factor of activated T cells) and CTLA-4[23,24], and posttranslational modifications including acetylation, phosphorylation, and ubiquitination are also indispensable for its regulatory function.

The expression of Foxp3 gene is controlled by both activation protein 1 (AP-1) complex, including c-Fos and c-Jun[25,26] and DNA methylation[27]. c-Fos and c-Jun interact with Foxp3 promoter and expedite Foxp3 expression. c-Fos knockdown reduced Foxp3 expression in $T_{reg}$[26]. Thus, c-Fos is a pivotal regulator associated with $T_{reg}$ activation. Furthermore, c-Fos also plays an important role in humoral responses. H2-c-Fos transgenic mice (overexpression of c-Fos) could constitutively express high levels of exogenous c-Fos in splenic T cells and B cells. Immunization of H2-c-Fos mice with antigens induced the production of primary immunoglobulin M (IgM) antibodies, but not that of IgG. Additionally, mice with overexpression of c-Fos were unable to generate splenic memory B cells, and the number of apoptotic germinal center B cells increased, suggesting that overexpression of c-Fos may induce germinal center B cell apoptosis[28]. Thus, it is postulated that c-Fos may exhibit positive and negative regulation on $T_{reg}$ and germinal center B cells.

In the current study, we primarily focused on a previously uncharacterized mechanism of the bilateral immunoregulation of DHA on $T_{regs}$ and circulating plasma cells. DHA directly interacts with the AP-1 complex, primarily the c-Fos transcription factor, enhances c-Fos activity in both $T_{reg}$ and plasma cells, but with opposite consequences. In $T_{reg}$, the activated c-Fos upregulated cell proliferation, whereas in circulating plasma cells, it promoted the expression of apoptosis-related genes and reduced cell population and immunoglobulin production.

## Results

**DHA reduced circulating B cell numbers but enhanced $T_{reg}$ expansion in peripheral blood.** Variations in total T and B lymphocytes, NK cells, CD4$^+$ and CD8$^+$ T cells, and active B cell subsets in the peripheral blood of mice after DHA treatment were first explored by flow cytometry. The number of total T lymphocytes and CD4$^+$ T cells in peripheral blood were markedly reduced (Fig. 1a, b), as were the absolute counts for total B lymphocytes, and active B cells, CD8$^+$ T cells, NK cells in the DHA-treated group compared to those in the control group (Fig. 1c–f). Moreover, the proportion of plasma cells secreting IgM and IgG was significantly lower in the peripheral blood of the DHA-treated group compared to that of the control group (Fig. 1g–j). However, the number of $T_{reg}$ in peripheral blood was found significantly increased in the DHA-treated group versus carboxymethyl cellulose solvent (CMC) control group (Fig. 1k, i). Thus, DHA suppressed B cell proliferation and enhanced $T_{reg}$ expansion.

**DHA upregulated genes associated with $T_{reg}$ expansion and circulating plasma cells apoptosis.** To further decipher the molecular mechanism underlying the effect of DHA on different immune cells in peripheral blood, we used single cell RNA sequencing (scRNA-seq) to assess differences in mRNA expression of cells and changes in immune cell subsets from the DHA-treated mice versus CMC-treated control mice. As scRNA-seq allows defining molecularly distinct cell subpopulations, we divided the immune cells into five populations based on commonly used immune cell mRNA markers. While DHA treatment caused a decrease in peripheral blood lymphocytes (including T cells, B cells, and NK cells), it increased the proportion of myeloid cells (including monocytes and peripheral granulocytes) in the blood (Fig. 2a–c and Supplementary Fig. 1). KEGG pathway analysis of the differentially expressed genes (DEGs) identified "drug metabolism-cytochrome P450" as the most significantly altered pathway in both peripheral monocytes and granulocytes (Fig. 2d). DHA treatment also upregulated VEGF and T-cell receptor signaling pathways, while decreasing NK cell-mediated cytotoxicity, as well as tyrosine metabolism and chemokine pathways in T cells (Fig. 2d).

A recent study has shown that immune suppression by $T_{reg}$ requires activation by IL-2 signaling[29]. Here, we found that IL-2 and interferon signaling were activated by DHA (Fig. 2f, g). Moreover, DHA consistently upregulated several genes associated with cell cycle progression, including mitochondria-related genes, ribosomal protein genes, and cyclin-dependent kinase (CDK), in $T_{reg}$ (Supplementary Data 1). DHA also inhibited DNA repair and oxidative phosphorylation (Fig. 2h–j), while upregulating the expression of apoptosis-related genes, including caspase-3 and Bax in the plasma cells (Supplementary Data 2), which is in consistent with the reduction of plasma cells (Fig. 1g–j). These results further explain the roles of DHA in up- and downregulation of $T_{reg}$ and plasma cells.

**DHA curbed lipopolysaccharide- and *Plasmodium berghei* ANKA-induced inflammation by reducing pro-inflammatory cytokines.** Many studies have revealed that bacterial lipopolysaccharides (LPS) can induce systemic inflammation[30,31]. As body

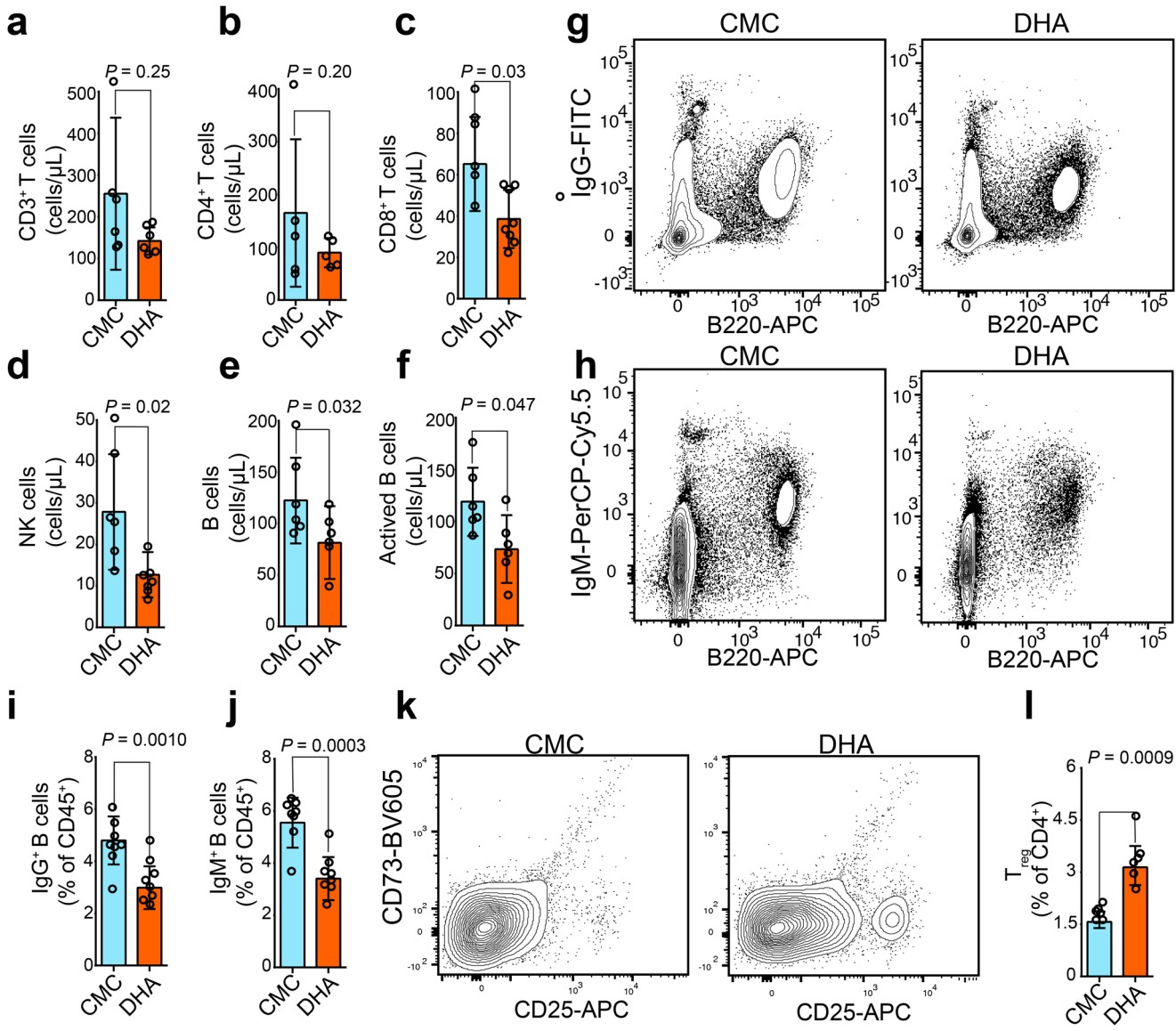

**Fig. 1 Dihydroartemisinin (DHA) suppressed plasma cells but promoted T_reg cell proliferation. a–f** Bar graphs represent decreased numbers of CD3+ T cells, CD4+ T cells, CD8+ T cells, NK cells, B cells, and activated B cells from DHA- and CMC-treated mouse groups (n = 5). **g, h** Representative images of B220+ IgG+ plasma cells and B220+ IgM+ plasma cells analyzed by cytometry. **i, j** Bar graphs represent significant decrease in percentages of B220+ IgG+ plasma cells and B220+ IgM+ plasma cells in DHA-treated group compared to CMC-treated group on day 26 post treatment (n = 8). **k** Representative flow cytometry images of T_reg cells from DHA- and CMC-treated mouse groups. **l** A bar graph represents significant increases in percentages of T_reg cells in DHA-treated mice versus CMC control group (n = 6). DHA = dihydroartemisinin group; CMC = carboxymethyl cellulose solvent solution control. The error bars indicate standard error. Statistical significance was determined with Mann–Whitney test.

weight is an important indicator of pathology severity following LPS challenge, we examined the effect of LPS on mouse body weight alterations in the context of DHA treatment. LPS-treated group exhibited significant weight loss compared to the control group on Day 5. Importantly, DHA administration significantly prevented LPS-induced body weight loss (Supplementary Fig. 2a). Additionally, our histological analysis revealed that DHA mitigated LPS-induced acute lung injury in mice compared to LPS group (Fig. 3a). These results indicated that DHA could alleviate LPS-induced systemic inflammation.

The immune suppressive effects of T_reg are achieved by secretion of soluble mediators and cytokines, such as IL10, IL-35, and TGF-β[32]. Previous report indicated that DHA induced T_reg activation, which is essential for the protection against LPS-mediated inflammatory injury[33]. Our result in this study showed that the proportion of CD152+ T_reg in the DHA-treated group was

significantly higher than that in the LPS- and un-treated groups (Fig. 3b). Moreover, the *P. berghei* ANKA-infected mice treated with LPS survived four days less than those only infected with *P. berghei* ANKA (Fig. 3c). On the contrary, DHA prolonged the survival time of LPS-treated mice previously infected by *P. berghei* ANKA (Fig. 3c). This aligned well with a previous finding that mice with pretreatment of DHA exhibited significantly decreased parasitemia and prolonged survival time[6]. We also observed that most parasites have been cleared upon DHA treatment (Supplementary Fig. 2b).

Inflammatory responses mediated by cytokines are presumably important in LPS or *P. berghei* ANKA-infected mice. It has been known that T helper 1 (Th1) cells produce interferon-γ (IFN-γ) and interlukin-2 (IL-2), whereas Th2 cells primarily produce IL-6[34], all of which play important roles in the regulation of innate immunity against pathogens. Meanwhile, tumor necrosis factor α (TNF α) is mainly secreted by activated macrophages and can

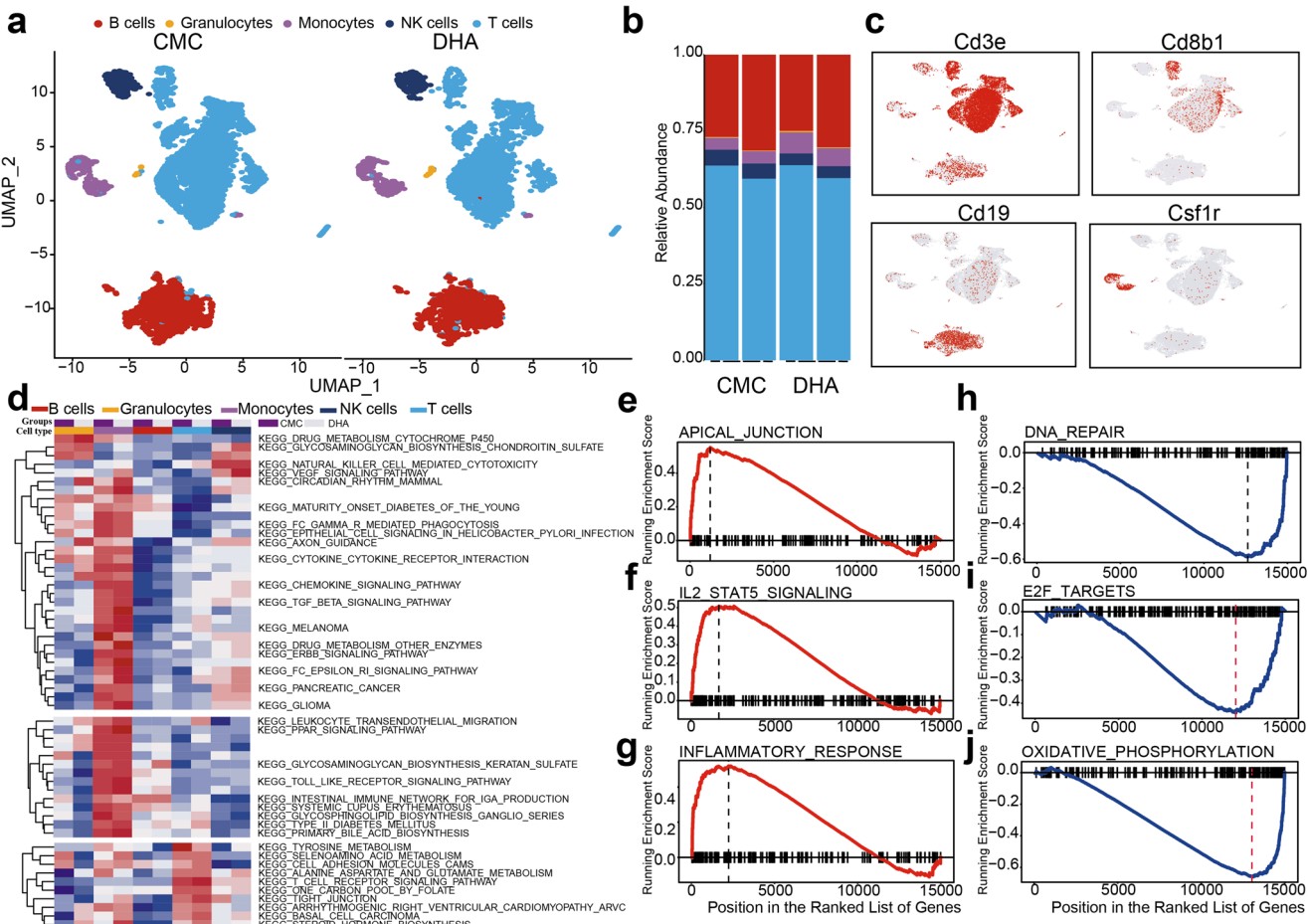

**Fig. 2 Single cell RNA sequencing revealed differential modulatory effects of dihydroartemisinin (DHA) on immune cells in peripheral blood. a** *t*-distributed stochastic neighbor embedding projections of immune cells in each group after biscuit normalization and imputation showing rich structure and diverse cell types. Cells color-coded by Biscuit clusters and labeled with inferred cell types. **b** Frequency maps (pileup maps) of cell-type fractions for each group's immune cells in peripheral blood, colored by cell type. **c** Immune cells were identified and annotated based on the automated SingleR analysis. *t*-distributed stochastic neighbor embedding projections of well-known marker genes including *CD3e* (T cells), *CD8b1* (T cells), *Cd19* (B cells) and *Csf1r* (monocytes). Red color indicated marker gene expressing cells. **d** Enriched KEGG pathway in DHA- versus CMC-treated group based on scRNA-seq. Changes of the pathways were standardized by Z scores. Colors represent original cell type and groups label as annotated. **e–j** The GSEA plot showed that DHA significantly upregulated apical junction, IL-2- stat5 and inflammatory response pathways, while downregulated DNA repairs, E2F targets, and oxidative phosphorylation pathways in DHA treated versus CMC control group.

effectively induce apoptosis and cell death. Additionally, IL-6 functions in the activation of humoral immunity, and $T_{reg}$ cells produce IL-10, thereby serving as "suppressor" T cells. We, therefore, quantified the abundance of various cytokines, namely, TNF α, IFN-γ, monocyte chemoattractant protein-1 (MCP-1), IL-2, IL-6 and IL-10 in the sera of mice in different treatment groups. As expected, the abundance of TNF α, IFN-γ, IL-2, IL-6 and MCP-1 was significantly elevated in the sera of the LPS-treated and *P. berghei* ANKA-infected groups compared to that of the healthy control group (Fig. 3d–h and Supplementary Fig. 2c–h). However, DHA treatment significantly dampened the responses of these pro-inflammatory cytokines (Fig. 3d–h). Importantly, IL-10, mainly functions in immune suppression, was downregulated in the DHA treatment group compared to that of the other groups (Fig. 3i). These results implied that DHA inhibited LPS- and *P. berghei* ANKA-induced inflammation (Fig. 3j).

**DHA inhibited the formation of splenic germinal centers (GCs) and immunoglobulin G (IgG)-producing plasma cells.** It is well known that B cell development is continued in spleen,

where mature naive B cells differentiate, in an antigen driven germinal center reaction, to memory B cells and plasma cells, both of which appear in the circulation. Many immunoglobulin G (IgG)-producing plasma cells are found in the germinal centers[35]. Here, the white pulp and GC area significantly shrank in DHA-treated mice compared to that of the control mice (Fig. 4a, b and Supplementary Fig. 3). Relative quantity of IgG in the sera of the mice also decreased on Day 8 and 15 after DHA treatment, compared to that of the control group (Fig. 4c, d). This result is in consistent with a decreasing in the absolute plasma cell number in peripheral blood (Fig. 1g–j).

To further confirm the impact of DHA on plasma cells, mice were experimentally immunized with sheep red blood cells (SRBC), and the GC response was analyzed 6 and 14 days post-immunization. At six days post-immunization, the proportion of splenic GC B cells in immunized mice were significantly increased compared to that of the non-immunized control mice. In contrast, the number of splenic GC B cells in immunized mice treated with DHA was significantly reduced compared to that in the control mice (Fig. 4e, f). Fourteen days post-immunization with SRBCs, the difference of splenic GC B cells between the

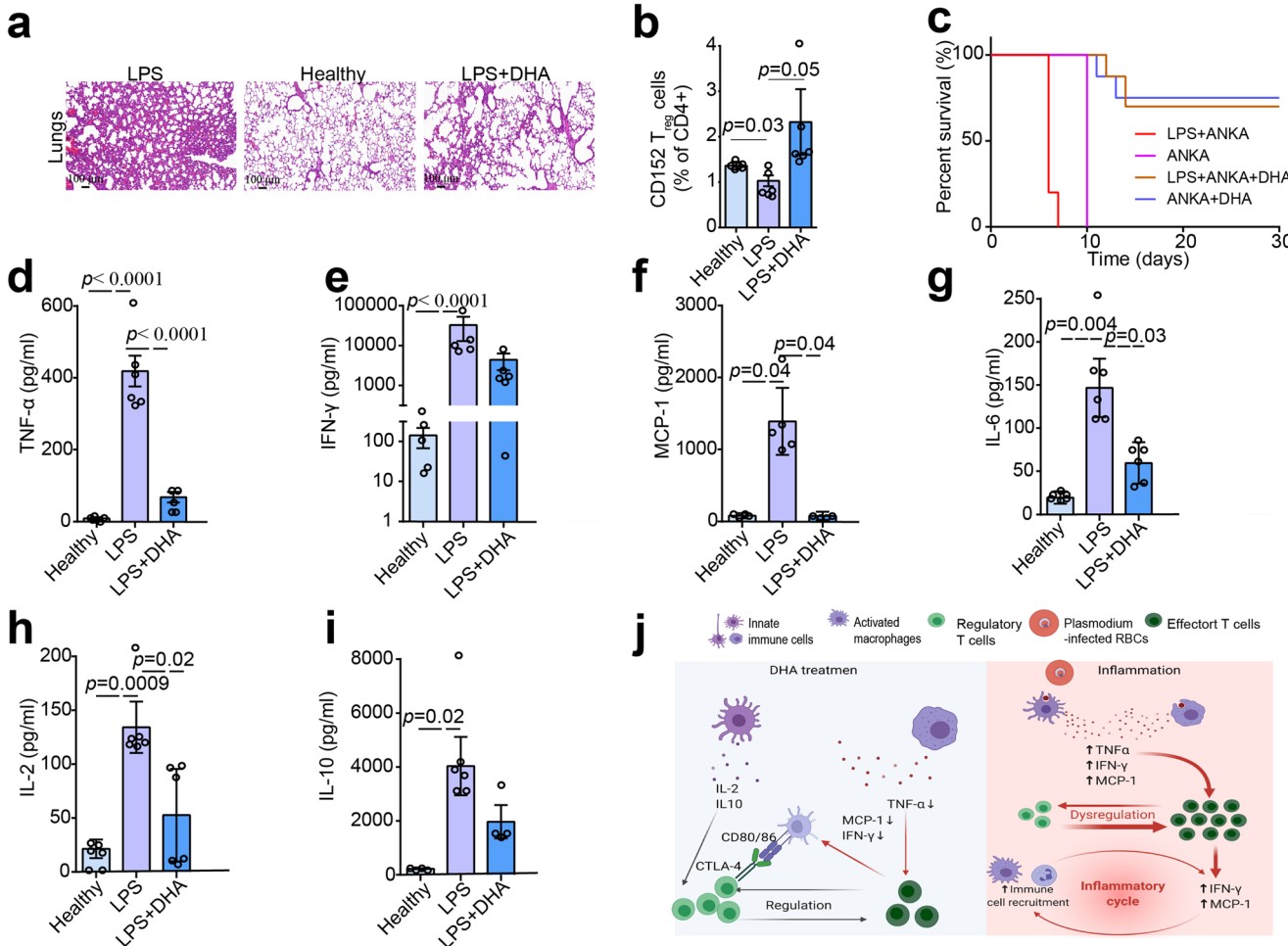

**Fig. 3 Dihydroartemisinin (DHA) alleviated the inflammation in mice induced by LPS and P. berghei infection. a** Representative sections of HE stained lung tissues of mice treated by LPS, DHA. LPS treatment increased the number of inflammatory cells in the alveolar spaces, destructed the pulmonary interstitium, and ruptured collagen fibers compared to that from the control group. DHA mitigated LPS-induced acute lung injury in mice. **b** Bar graphs represent significant decrease in percentages of $T_{reg}$ in LPS treated group compared to healthy group, while significant increase in percentages of $T_{reg}$ in DHA treated group compared to LPS treated group. Proportion of total CD152+ $T_{reg}$ gated off the total CD4+ populations ($n = 6$, two-tailed Student $t$-test). **c** Survival curve of mice following single *P. berghei* ANKA infection or combined with LPS treatment under DHA treatment or without DHA treatment. The *P. berghei* ANKA-infected mice treated with LPS survived four days less than those only infected with *P. berghei* ANKA. On the contrary, DHA prolonged the survival time of *P. berghei* ANKA-infected mice treated with LPS ($n = 8$–10). Mice injected with LPS were previously infected with *P. berghei* ANKA for 4 days in LPS + ANKA group. **d**–**i** Serum abundance of IFN-γ, tumor necrosis factor α (TNFα), IL-2, MCP-1, IL-6, and IL-10 increased in LPS-treated mice compared with healthy mice while significantly decreased in DHA-treated mice ($n = 4$–6, two-tailed Student t-test). The sera were collected at day 7 after DHA treatment and the levels of the cytokines were determined by cytometric bead array (CBA). **j** A schematic diagram of DHA in its modulation on immune cells. DHA inhibited LPS- and *P. berghei* ANKA- induced inflammation. Figure was created using BioRender and Agreement number was GC24OV5V07. The error bars indicate standard error.

DHA-treated and untreated mice were not significant (Supplementary Fig. 4a, b).

Furthermore, the total splenic plasma cells (PC B cells) were found significantly decreased in the DHA-treated group compared to that in the control group at day 6 and 14 post-immunization with SRBCs (Fig. 4g, h). Although the splenic IgG-producing PC B cells were not significantly decreased on Day 6 post-immunization, a significant decrease was observed 14 days post-immunization under DHA treatment (Fig. 4i, j). In agreement with the lower frequency of IgG-producing PC B cells in the spleen of DHA-treated mice, significantly lower levels of IgG were detected in the sera of DHA-treated mice compared to that of the control mice (Fig. 4k). Thus, DHA treatment suppressed GC formation, reduced the plasma cells population, and inhibited IgG production, suggesting that it may curtail humoral immunity in mice (Fig. 4l).

**DHA upregulated *c-Fos* expression in plasma cells and $T_{reg}$ cells**. To further decipher the mechanism underlying the spontaneous regulation of DHA on both $T_{reg}$ and B cells, we applied single cell regulatory network inference and clustering (SCENIC) to establish gene regulatory networks of different T and B cell subsets in response to DHA treatment. Moreover, we found that $T_{reg}$ exhibited highly transcriptional activities of *c-Fos*, *Cebpb*, *Cux1*, *Runx1*, and *Nr3c1* in DHA group compared to that of the CMC control group (Fig. 5a). Meanwhile, the transcriptional activities of *c-Fos*, a transcription factor associated with the MAPK pathway, was also significantly higher in the PC B cells in DHA group than that in the CMC control group. In addition, we observed that the transcription of apoptosis-related transcription factor Klf4 was upregulated in the DHA group compared to that in the CMC group (Fig. 5b). To further investigate the key transcription factors that control the bilateral immunomodulation of DHA, we analyzed the

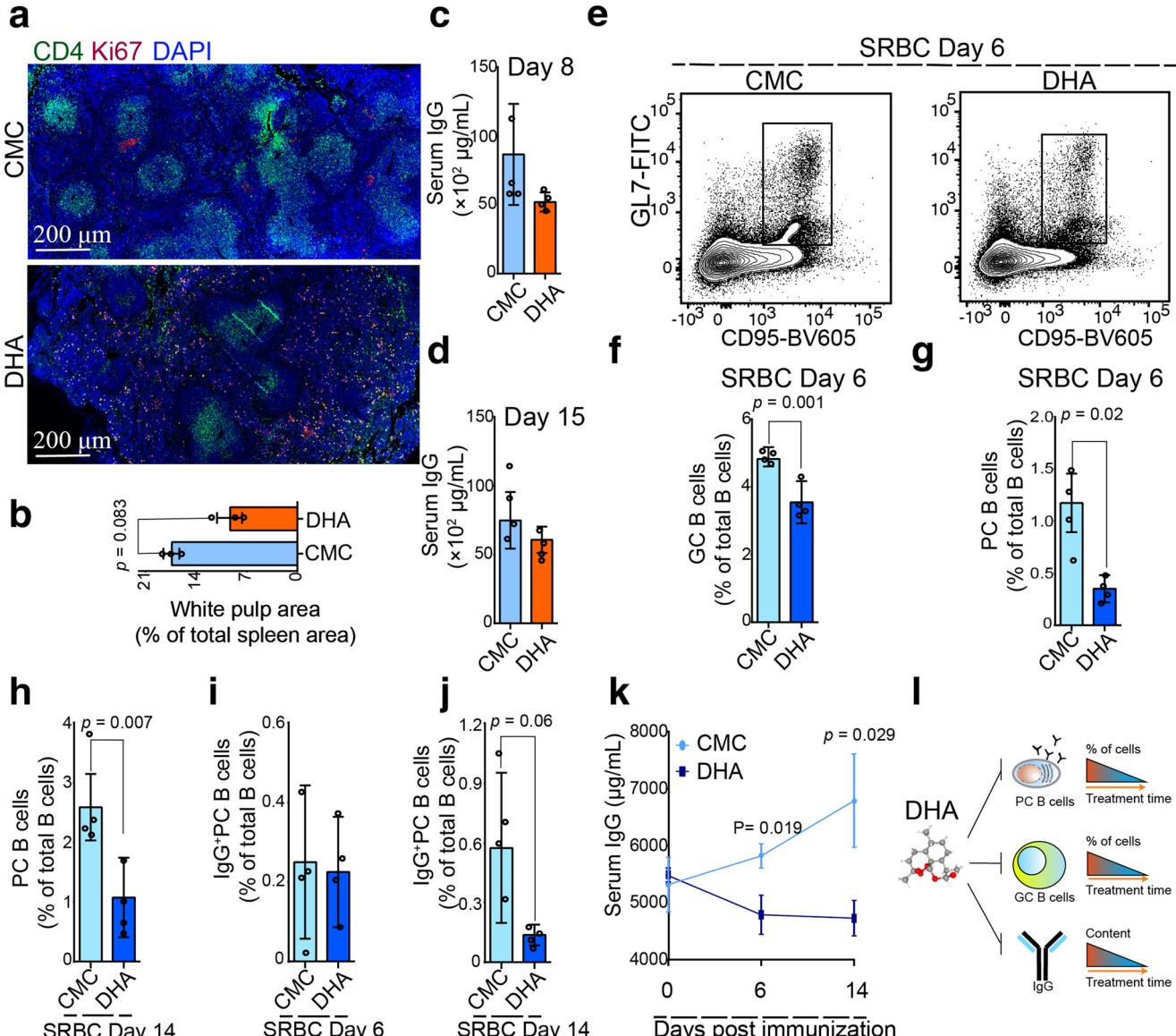

**Fig. 4 Dihydroartemisinin (DHA) suppressed humoral immune responses in sheep red blood cell-immunized mice. a** Representative spleen sections of mice treated with DHA and CMC control. In the white pulp (green), germinal center B cells were highly proliferative and expressed a gene known as $Ki67$ (red). Splenic CD4[+] T cells (green), germinal center B cells (red) were marked with anti-CD4 and anti-Ki67 antibodies. **b** A bar graphs represent significant decrease in area of white pulp in DHA-treated group compared to CMC-treated group on day 26 post treatment ($n = 3$, two-tailed Student t-test). **c**, **d** Bar graphs represent a marked decrease in serum total IgG after 6 and 15 days treatment with either DHA or CMC ($n = 3$, two-tailed Student t-test). **e** Representative flow cytometry plots of germinal center B cells (B220[+]CD95[+]GL7[+]) from DHA- and CMC control mice. **f** Bar graphs represent significant decreases of germinal center B cells (B220[+]CD95[+]GL7[+]) and in DHA-treated versus CMC-treat group post immunized with sheep red blood cells at 6 days ($n = 4$, two-tailed Student t-test). **g–j** The total splenic plasma cells (PC B cells) were significantly decreased in the DHA-treated group compared to that in control group at days 6 and 14 post-immunization with SRBC ($n = 4$, two-tailed Student t-test). **k** Serum anti-SRBC IgG antibodies were quantitated by ELISA at days 0, 6, and 14 post immunization ($n = 5$). **l** A schematic representation of how DHA suppressed humoral immune responses. The error bars indicate standard error.

intersection of these factors across different cell populations. Among the 10 top transcription factors, *c-Fos* was the only transcription factor that was upregulated in both T$_{reg}$ and B cells (Fig. 5c). To analyze the signaling pathways regulated by *c-Fos*, we used gene set variation analysis (GSVA) to enrich the differentially expressed genes in PC B cells. These results indicated that DHA significantly reduced PC B cell activation (Fig. 5d, e). Thus, DHA likely exerted spontaneous regulation on both T$_{reg}$ and PC B cells.

We then performed spatial reconstruction of PC B cells in vivo and profiled c-Fos expression. Interestingly, in the control group, of the Ki67[+] PC B cells located in GCs, only a few cells expressed

c-Fos (Fig. 5f). In contrast, Ki67[+] PC B cells disappeared from the GCs of the DHA group, and the remained cells expressed more c-Fos (Fig. 5f).

**c-Fos inhibitor counteracted gene activation by DHA in plasma cells and T$_{reg}$ cells.** To further verify the positive and negative regulation of c-Fos on T$_{reg}$ and plasma cells, we applied c-Fos specific inhibitor (T5224) in combination with DHA treatment, as T5224 can efficiently inhibit the leucine zipper domain of c-Fos but not impacts other transcription factors. Firstly, the expression of c-Fos was

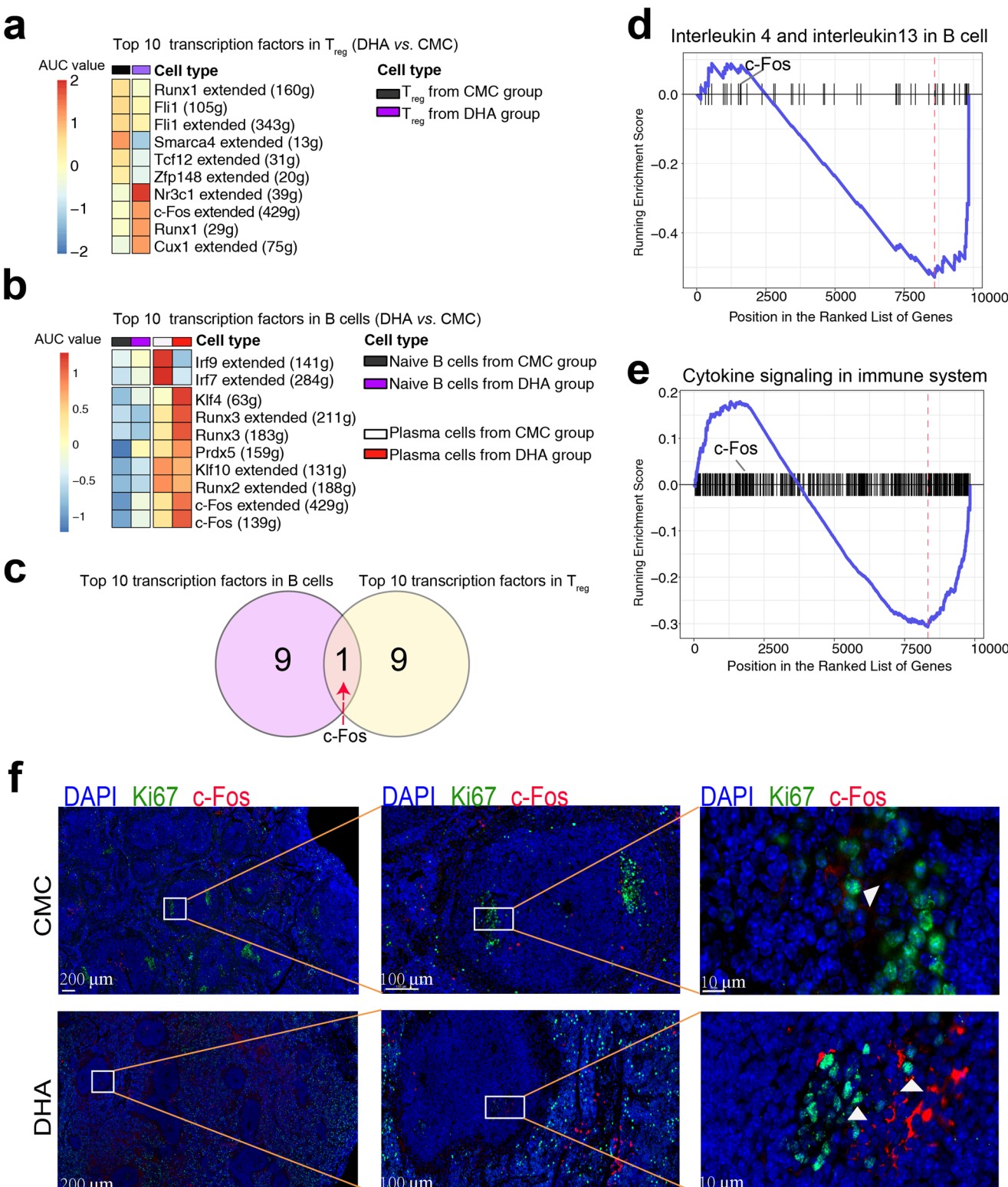

**Fig. 5 Dihydroartemisinin (DHA) upregulated c-Fos transcription and expression in T_reg and PC B cells. a**, **b** The differential transcription activity of the top 10 transcription factors in T_reg and B cells by SCENIC. Color scaling represents the normalized AUC values of target genes in the regulon being expressed as computed by SCENIC. **c** Of the 10 top transcription factors in T_reg and B cells, *c-Fos* was the only transcription factor that was upregulated in both T_reg and B cells by DHA. **d**, **e** GSVA analysis of plasma cells indicated that the plasma cells from DHA treated mice had significantly lower expression of B cell activating associated genes that were regulated by c-Fos than those from CMC control. **f** Representative immunofluorescence images of spleen sections from DHA treated mice and CMC control mice stained with anti-Ki67 antibodies (green) and anti-c-Fos antibodies (red). Cell nuclei were stained with DAPI (blue). DHA treatment led to elevated expression of c-Fos in plasma cells, and atrophy of germinal centers. Fos = c-Fos.

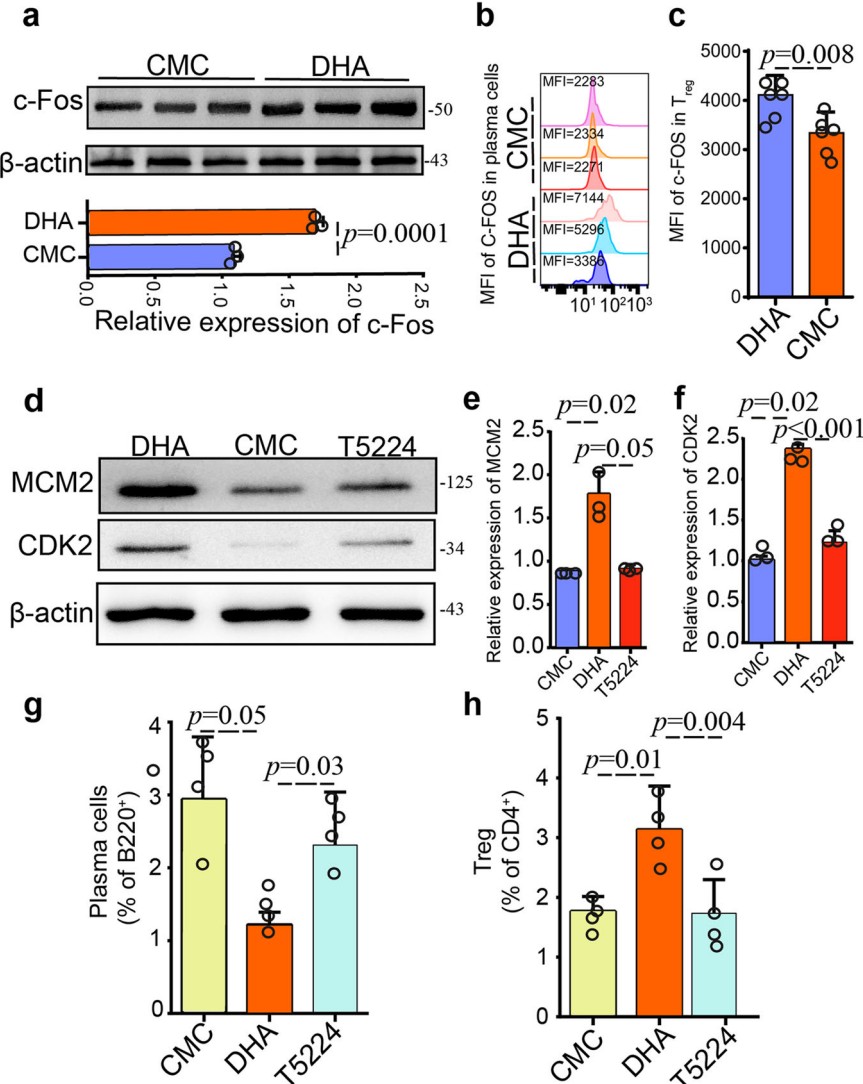

**Fig. 6 Dihydroartemisinin (DHA) induced c-Fos expression in both T$_{reg}$ and PC B cells. a** The expression variations of c-Fos in peripheral blood were determined by Western blot, with β-actin serving as a loading control. Bar graph shows differences in normalized abundance of the c-Fos between CMC control and DHA-treated groups ($n = 3$, Student t-test). **b**, **c** Quantification of the MFI of the c-Fos transcription factors in plasma cells and T$_{regs}$ from CMC and DHA treated mice. Δ MFI (Mean Fluorescence Intensity) = MFI antibody – MFI isotype control. DHA-treated mice express more c-Fos than CMC-treated mice in T$_{regs}$ and plasma cells ($n = 6$, Student t-test). **d** Western blot analysis of CDK2 and MCM2 proteins in peripheral blood T$_{reg}$ from mouse blood treated with DHA alone, DHA with the T5224 inhibitor, and CMC control was performed with protein specific antibodies. **e**, **f** Bar graph shows differences in normalized abundance of the CDK2 and MCM2 between CMC control, DHA treatment, and DHA plus T5224 groups ($n = 3$, Student t-test). Increased CDK2 and MCM2 expression were observed in T$_{reg}$ treated with DHA compared to that of CMC-treated mice, whereas CDK2 and MCM2 expression were greatly reduced in mice treated with T5224 in the presence of DHA. **g**, **h** Bar graph shows differences in percentage of circulating plasma cells (B220-gated) and peripheral blood T$_{reg}$ (CD4-gated) in the CMC alone, DHA and DHA plus T5224 treated groups ($n = 4$, Student t-test). The error bars indicate standard error.

significantly upregulated (Fig. 6a), which was in consistence with the SCENIC results. To further validate and confirm the single-cell sequencing results that c-Fos was up-regulated in both plasma cells and T$_{reg}$ cells, c-Fos protein in the two cell types was assayed by flow cytometry. These results were in line with that from the single cell sequencing, suggesting that c-Fos expression was really upregulated in plasma cells and T$_{regs}$ by DHA (Fig. 6b, c). Additionally, the expression of the proliferation marker CDK2 and MCM2 in T$_{reg}$ was increased in mice treated with DHA compared to that of CMC-treated mice, whereas CDK2 and MCM2 expression were greatly reduced in mice treated with DHA in the presence of T5224 (Fig. 6d–f), implying that CDK2 and MCM2 expression were regulated by c-Fos. Importantly, T5224 significantly inhibited the regulatory effect of DHA on both plasma cells and T$_{reg}$ cells (Fig. 6g, h).

**DHA directly interacted with the c-FOS protein**. The direct interaction of DHA with c-Fos protein was confirmed with Bio-layer interferometry (BLI). DHA exhibited high binding affinity to murine c-Fos (KD (M) = 0.001194, KD Error= 0.0009775), but it did not bind to an irrelevant his-tagged control protein (Fig. 7a, b).

**DHA enhanced the binding of c-Fos on target genes in the recognition sites**. To decipher the c-Fos-DHA interaction further, we predicted the putative DHA-binding domain of both murine c-Fos (2.3 Å, PDB accession number 2WT7) and human counterpart (3.05 Å, PDB accession number 1FOS) by docking simulation. c-Fos, formed a single helix but constituted a heterodimer fork structure with murine MafB or human c-Jun[36]. and DHA was docked preferentially at the N-terminus of the basic DNA binding

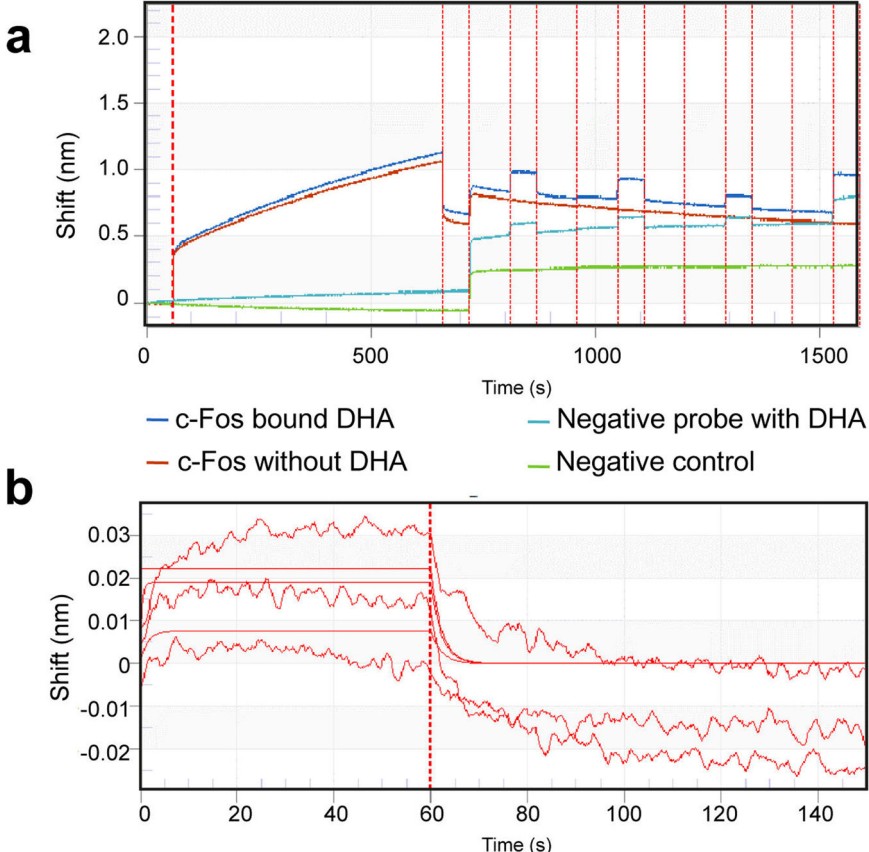

**Fig. 7 DHA interact directly with c-Fos. a** The binding of DHA to immobilized recombinant c-Fos at 50 ng/mL was determined by biolayer interferometry. Blank probe (Green), blank probe with only DHA (Sky blue), and c-Fos- immobilized probe (Red) without as negative controls. **b** Binding curves showing association and dissociation kinetics for each concentration of DHA to c-Fos pair.

region of the fork structure to form hydrogen bonds with the Arg 158 amino acid (−8.9 kcal/mol, Supplementary Fig. 5a), resulting in the formation of a stable complex, in the murine c-Fos with or without DNA (Fig. 8a–d). Importantly, the binding of c-Fos to DNA (the fifth base G) increased significantly with the addition of DHA (Fig. 8c, d), implying that DHA may enhance AP-1 trans-activation efficiency. While in human cells, it seemed DHA could bind to c-Jun, but not c-Fos in the non-native states (c-Fos-c-Jun complex without DNA) (Fig. 8e, f). However, in the native states, DHA formed hydrogen bonds with the 158th Arginine of c-Fos and the thirty-first adenine (A) of the recognized DNA (Fig. 8g, h, Supplementary Fig. 5b, and ref. [36]) in a similar way as observed with murine c-Fos (Fig. 8a–d).

## Discussion

Dihydroartemisinin (DHA) is not only a frontline drug for the treatment of malaria, but also exhibits potent therapeutic implication in certain autoimmune diseases[9]. Its anti-malaria efficacy is believed to be involved in the depolarization of the parasite mito-chondrial, plasma membranes and heme-mediated degradation pathways[37], whereas the mechanism of its curative effect on immune cells is still largely elusive. Deep understanding the action mode of DHA on the immune cells would further justify the proposition of its expansion in medical applications. Here, we reported that DHA increased the proportion of $T_{reg}$ while decreased the number of plasma cells both in healthy, LPS- and *P. berghei* ANKA infected mice and SRBC immunization model (Figs. 1–5). In addition, $T_{reg}$ activated by DHA overcome inflammation induced by LPS- and *P. berghei* ANKA infection (Fig. 3 and Supplementary Fig. 2c–h). DHA treatment also suppressed the

anti-SRBC IgG response in vivo (Fig. 4). We further revealed that DHA upregulated the expression of cell proliferation-associated genes in $T_{reg}$ and apoptosis-related genes in plasma B cells by promotion of c-Fos expression (Figs. 5 and 6).

The numbers of peripheral lymphocyte populations are considered an important indicator in determining the status of the immune system. Patients with systemic lupus erythematosus frequently have elevated numbers of total T and B cells in the peripheral blood, but the $T_{reg}$ cells are significantly reduced, which companies with increased immunoglobulins and disease severity[38]. DHA is one of the few successful therapeutic agents proposed for systemic lupus erythematosus, but its potential mechanism remains poorly understood. In the current study, we observed a significant reduction in the number of CD8[+] T cells, NK T cells, total B cells, and activated B cells, but significant increase in the $T_{reg}$ population in the peripheral blood following DHA treatment (Fig. 1). ScRNA-seq analysis revealed that the genes controlling cell proliferation in $T_{reg}$ were increased; in contrast, the genes associated with apoptosis in plasma cells was promoted, which supported the distinct roles of DHA in the regulation of $T_{reg}$ proliferation and plasma cell reduction (Figs. 1 and 2).

The bilateral immunomodulatory effects of DHA were further evaluated using three classical immunological models including LPS-induced inflammation model, experimental malaria models and SRBC immunization model. Mice with systemic inflammation induced by LPS have been frequently used in testing drugs with function of potential $T_{reg}$ cells activation[39]. Here, we found that DHA inhibited the inflammation inducted by both LPS and *P. berghei* infection, and all were due to the activation of

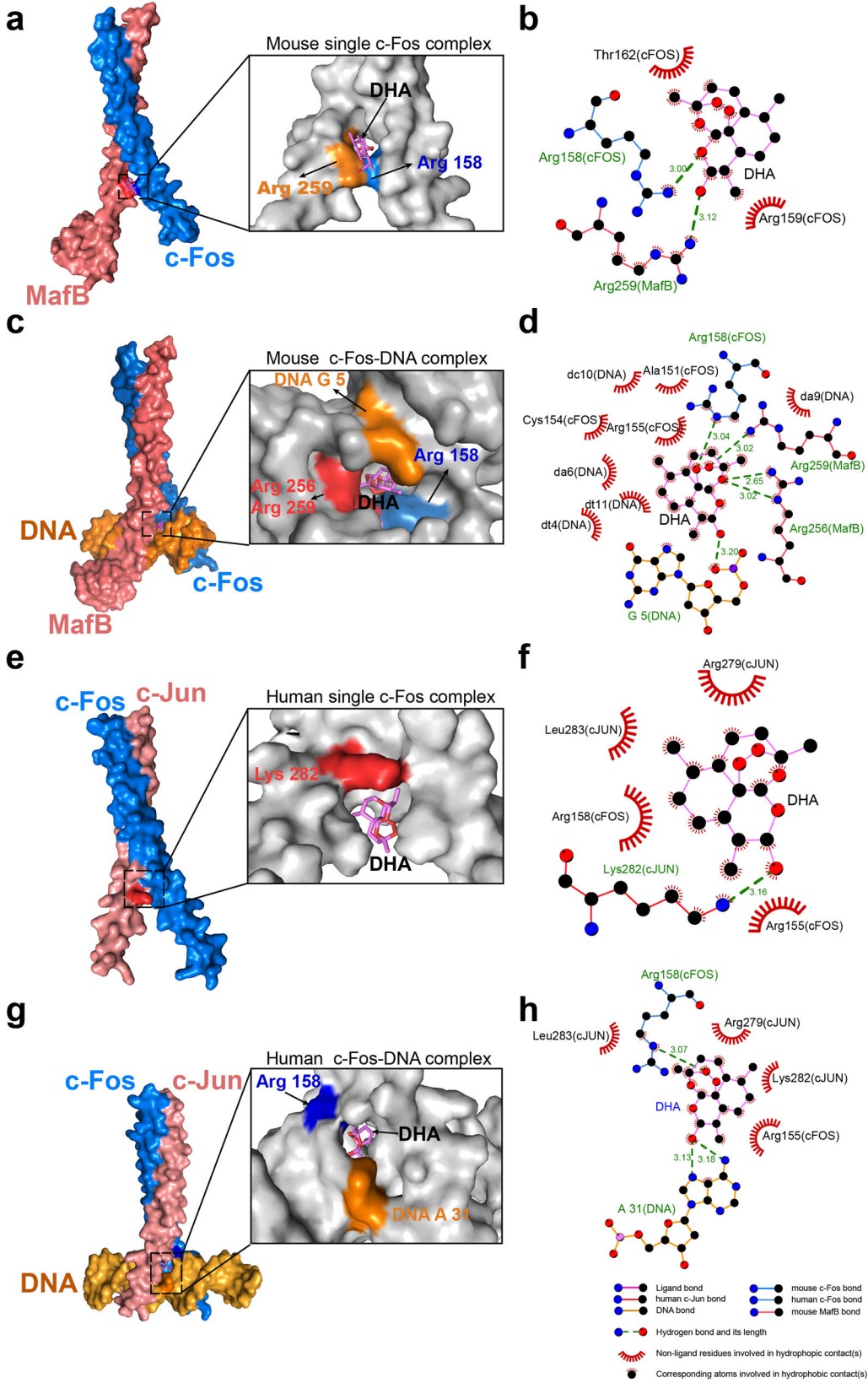

T_reg cells. Furthermore, we found that DHA reduced inflammatory cytokines (including IL-6, TNF α, IFN-γ, and MCP-1) in mice either treated with LPS or infected with the malarial parasite *P. berghei* (Fig. 3). The data were in line with a previous finding that DHA effectively inhibited TNF-α expression in peritoneal macrophages of BXSB mice[40].

Furthermore, with a mouse model of SRBC immunization, which has been frequently applied in study on GC reactions in the context of immunomodulation[41], we found that treatment with DHA significantly decreased the humoral response in SRBC-immunized mice, but promoted T_reg expansion, DHA exhibited efficient inhibition on humoral immunity (Fig. 4). Thus, it can be

**Fig. 8 DHA binds c-Fos and zipped DNA in the transcription fork. a** Schematic of the docking interactions between murine c-Fos complex without DNA (PDB code: 2WT7) and DHA. DHA binds murine c-Fos protein, stabilize the interaction by forming a hydrogen bond with Arg 158. **b** The interaction between the murine c-Fos and DHA (3D) without DNA. **c** Schematic of the interactions between murine c-Fos and DHA with DNA complex (PDB code: 2WT7). DHA not only binds to mouse c-Fos protein, stabilizes the interaction by forming a hydrogen bond with the 158th Arg but also binds to DNA (the 5th G). **d** The interaction between murine c-Fos with DNA complex in the presence of DHA (3D). **e** Schematic of the docking interactions between human c-Fos complex without DNA (PDB code: 1FOS) and DHA. DHA binds c-Jun protein, stabilizes the interaction by forming a hydrogen bond with Lys 282. **f** The interaction between human c-Fos complex and DHA without DNA (3D). **g** Schematic of the docking interactions between human c-Fos-c-Jun with DNA complex (PDB code: 1FOS) and DHA. DHA not only binds to human c-Fos protein, stabilizes the interaction by forming a hydrogen bond with the 158th Arg residue but also binds to DNA at the thirty-first adenine position. **h** The interaction between human c-Fos-c-Jun with DNA complex and DHA (3D). The error bars indicate standard error.

concluded that DHA can spontaneously activate $T_{reg}$ cells and suppress plasma cells.

The molecular mechanism of DHA modulation of the immune system was deeply dissected. ScRNA-seq revealed predominant transcription of *c-Fos* in both $T_{reg}$ and circulating plasma cells in DHA-treated mice. c-Fos transcription factors have been known to regulate cell proliferation, survival, and apoptosis[42]. A previously study showed that c-Fos could promote *SIRT6* and *NF-κB* expression by suppression of histone H3K9 acetylation, and consequently induced apoptosis of tumorous hepatic cells[43,44]. In contrast, the activation of c-Fos also induces *Foxp3* expression and prevents the development of autoimmune diseases[26]. Here, we found that the expression of c-Fos was promoted by DHA in both $T_{reg}$ and circulating plasma cells by DHA (Figs. 5–8). DHA induced significant overexpression of proliferation-related genes, which are regulated by c-Fos, in $T_{reg}$ cells. However, DHA also significantly induced high expression of c-Fos and activation of apoptosis-related genes in splenic GC B cells[6] and circulating plasma cells, which resulted in the reduction of plasma cells in the spleen and peripheral blood.

One of the conclusions of this work is that DHA can upregulate c-Fos expression and enhances its interaction with target genes in both $T_{reg}$ and B cells with bilateral cell fates. Importantly, the bilateral function and of DHA-induced c-Fos in $T_{reg}$ and plasma cells were further verified with a specific inhibitor (T5224) which efficiently inhibits the leucine zipper domain of c-Fos but not impacts other transcription factors. The effect of DHA on $T_{reg}$ proliferation and plasma cells apoptosis were inhibited by T5224 (Fig. 6), which supported its distinct role in the specific activation of c-Fos in both $T_{reg}$ cells and B cells. Since the spleen and peripheral blood are highly complicated niches for cross-talk of multifarious immune cells, we can rule out the roles of other immune cells which might participate in the regulation.

Importantly, to the best of our knowledge, we demonstrate here that DHA directly bound both human and murine c-Fos on the DNA binding domain, forming a more stable c-Fos-DNA complex, which facilitates the expression of downstream genes in both $T_{reg}$ and plasma cells (Fig. 9). To the best of our knowledge, this study is the first to show the association between DHA and c-Fos on the DNA binding domain. Furthermore, our study indicated that it is more important to monitor changes in immunologic indicators in malaria patients during prolonged treatment with DHA.

Taken together, we revealed that DHA can spontaneously activated c-Fos transcription factor in both $T_{reg}$ and circulating plasma cells, but with opposite consequences. In $T_{reg}$, the activated c-Fos upregulated cell proliferation, whereas in circulating plasma cells, it promotes the expression of apoptosis-related genes and reduces plasma cells and immunoglobulin production.

## Methods

**Mice**. Female BALB/c and C57BL/6 mice (approximately 18–22 g) were obtained from Liaoning Changsheng Biological Technology Company (Liaoning, China).

The animal experiments were conducted according to the animal husbandry guidelines of Shenyang Agricultural University (permit no. SYXK < Liao>2021-0010).

**Preparation of DHA solution**. The 0.1 mg/mL DHA solution was prepared as follow[7]. Briefly, in 20 mL of distilled water, 0.1 g of CMC (Solarbio Company, Beijing, China, Catalog No. 9004-32-4) was dissolved and sequentially added 0.2 g DHA (Puyi Biological Company, Nanjing, China; Catalog No. PY1835126Q). CMC without DHA was used as a solvent control.

**Analysis of immunological regulation of DHA in healthy BALB/c mice**. To probe immunomodulatory activity, DHA or its solvent was intragastrically administered to healthy mice. Female BALB/c mice were randomized into two groups and administered 200 μL of 0.1 mg/mL DHA or their solvent control 200 μL of a 0.5% CMC once per day until 26 days. At 26 days, the mice were euthanized and splenic tissues and blood were collected for immunological assessment.

**Analysis of the suppressive effect of DHA on systemic inflammation induced by low-dose LPS in a mouse model**. Systemic inflammation was induced by intraperitoneal injection of LPS (Sigma-Aldrich) to female BALB/c mice at a dose of 5 mg/kg for three consecutive days. One hour after LPS injection, mice were intragastrically administered 200 μL of 0.1 mg/mL DHA solution once per day or the control CMC solution. Seven days post-administration, the total $T_{reg}$ population in peripheral blood was analyzed by flow cytometry. Absolute cell numbers were quantified by flow cytometry using counting beads. $T_{reg}$ were identified as CD4 $^+$ CD25 $^{+/hi}$ CTLA-4$^{hi}$ cells.

**Analysis of the suppressive effect of DHA on inflammation induced by *Plasmodium berghei* ANKA strain infection in combination with LPS**. In an experimental murine malaria model, $10^3$ *P. berghei* ANKA-infected blood cells were injected intraperitoneally into each mouse in the four groups. The DHA group received DHA alone; the CMC group received CMC alone; the LPS group received LPS (5 mg/kg) i.p. after *P. berghei* ANKA-infection; the LPS + DHA group received 200 μL of 0.1 mg/mL DHA i.g. plus LPS (5 mg/kg) i.p. after *P. berghei* ANKA infection. DHA administration was initiated at day 4 post-infection. Mice injected with LPS were previously infected with *P. berghei* ANKA for 4 days in LPS + ANKA group. Parasitemia was determined using Giemsa-stained blood smears every two days post-infection.

**Analysis of DHA on the inhibitory effect on humeral responses in murine model of immunization with sheep red blood cells (SRBC)**. SRBC (10%) were purchased from NanJing SenBeiJia Biotechnology Co., Ltd. and resuspended at $10^7$ cells/mL, followed by intraperitoneal injection of 200 μL into adult C57BL/6 mice. Following SRBC injection, mice in DHA group and CMC group were continuously gavaged with 100 mg/kg DHA (200 μL) or equal volume of CMC solution until the end of the experiment. On Day 6 or 14 post-immunization, total splenic PC B cells, IgG producing PC B cell, or GC B cells were enumerated by flow cytometry using a BD FACS Aria flow cytometry system (BD Biosciences, San Jose, CA, USA).

**Flow cytometric detection of immune cells in the peripheral blood**. Following pre-incubation with a purified anti-mouse CD16/32 antibody, the cells were then incubated with specific antibodies or isotype controls, according to the manufacturer's guidelines. GC B cell polarization was evaluated by staining with anti-B220-APC (clone RA3-6B2, 0.2 mg/ml), anti-CD19-PECy7 (clone 6D5, 0.2 mg/ml), anti-CD38-Alexa Fluor 700 (clone 90, 0.2 mg/ml), anti-CD95-BV605 (clone DX2, 0.2 mg/ml), and anti-GL7-FITC (clone GL7BD, 0.2 mg/ml; BioLegend). Plasma cell B cell polarization was evaluated by staining with anti-B220-APC (clone RA3-6B2, 0.2 mg/ml), anti-CD19-PECy7 (clone 6D5, 0.2 mg/ml), anti-CD38-Alexa Fluor 700 (clone 90, 0.2 mg/ml), anti-CD138-BV421 (clone 281-2, 0.2 mg/ml), anti-CD267/TCAI-PE (clone 8F10, 0.2 mg/ml), anti-IgM-Percp/cy5.5 (clone RMM-1, 0.2 mg/ml), and anti-IgG-FITC (clone Poly4060, 0.2 mg/ml, BioLegend). For intracellular staining, cells from peripheral blood were fixed with freshly prepared True-Nuclear™ Transcription Factor Buffer Set (BioLegend), washed twice with True-Nuclear™ Perm Buffer (BioLegend),

**Fig. 9 A schematic summary of dihydroartemisinin (DHA)-mediated positive and negative regulation on T$_{reg}$ and plasma cells.** DHA promoted T$_{reg}$ activation, but suppressed plasma cells and suppressed IgG production. The differential regulation maneuver was through the direct activation of c-Fos by DHA, which induces genes of cell proliferation in T$_{reg}$ and genes of cell apoptosis in plasma cells. Figure was created using BioRender and Agreement number was AZ23VK4EWZ.

stained with primary antibody at 4 °C for 1 h (anti-c-fos, Affinity), washed twice with True-Nuclear$^{TM}$ Perm Buffer, stained with secondary antibody at room temp for 1 h (goat anti rabbit Alexa Fluor 488, Thermo Fisher Scientific), and washed twice again with True-Nuclear$^{TM}$ Perm Buffer. The MFI of c-fos in T$_{regs}$ was gated on zombie$^-$CD45$^+$ CD4$^+$CD25$^+$Foxp3$^+$ and the MFI of c-fos in plasma cells was gated on zombie$^-$CD45$^+$ CD19$^+$B220$^+$ CD38$^+$ CD138$^+$ (Supplementary Fig. 6). Immune cells were detected and analyzed using a fluorescence-activated cell sorting Aria III flow cytometer (BD Biosciences, San Jose, CA, USA), and the gates were defined using the isotype and fluorescence minus one control.

**Immunofluorescence histochemistry.** For immunofluorescence histochemistry analysis of PC B cells in the spleen of mice treated with DHA, paraffin-embedded spleen tissue sections were rehydrated and underwent antigen retrieval, followed by antigen blocking with PBS solution containing 4% normal goat serum and the splenic white pulp was visualized by indirect immunofluorescence after staining with a mouse anti-CD4 primary monoclonal antibody (1 mg/ml) and FITC-conjugated goat anti-mouse secondary antibody (Thermo, 2 mg/ml). The germinal center B cells were visualized by indirect immunofluorescence after staining with a rabbit anti-Ki67 primary antibody (1 mg/ml) and 594-alexa-conjugated goat anti-rabbit secondary antibody (Thermo, 2 mg/ml). The expression of c-Fos in germinal center was detected using mouse anti-c-Fos primary antibody with a 594-alexa-conjugated goat anti-mouse secondary antibody. The signals were recorded and analyzed by Pannoramic SCAN and Pannoramic Viewer (3D HISTECH), respectively.

**In vivo c-Fos inhibitor experiments.** Further confirming the effect of DHA on c-FOS activity, the mice of DHA plus T5224 group were given T5224 (30 mg kg$^{-1}$) and DHA (200 μL, 0.1 mg/mL) orally for 8 consecutive days. Mice in the control group received DHA alone, CMC, and an equal volume of solvent control at the dosage recommended by the manufacturer. T$_{reg}$ cells were isolated from the mice and subsequently lysed and extracted by using total protein extraction kit. The precipitated protein from T$_{reg}$ was dissolved in sample buffer and separated with 10% SDS-PAGE and subsequently was transferred to PVDF membranes. After transferring to PVDF membranes, proteins were detected using MCM2- or CDK2-

or β-actin-specific antibodies. T5224 was purchased from MedChemExpress Company.

**Enzyme linked immunosorbent assay.** Total IgG concentrations in the sera of mice treated with DHA or CMC control in healthy or SRBC- immunized groups were detected using a mouse IgG ELISA kit (Elabscience, Wuhan, China; Cat: E-TSEL-M0003). Briefly, an ELISA plates were coated with anti-mouse IgG antibodies to capture serum IgG, and blocked with PBS with 5% skimmed milk powder. Serum samples (50 μL; 1:10,000) in each group were added to the wells and incubated with 50 μL of biotinylated anti-mouse IgG and then incubated for 90 min at 37 °C in an incubator. Plates were then washed four times with PBST (PBS with Tween-20) buffer. After washing, 100 μL HRP conjugate substrate working solution was added to the well and incubated for 30 min. Then the plates were washed with PBST four times and the color reaction was initiated by addition of 50 μL TMB (Tetramethyl Benzidine) solution and then incubated for 15 min at 37 °C in an incubator. The absorbance at 450 nm was measured and analyzed by using a microplate reader. The ELISA experiment was performed in duplicate, with at least six independent experiments.

**Detection of cytokines in DHA-treated and the control mice.** For CBA multiplex cytokine detection, sera were collected from DHA- and CMC control mice, and cytokines were determined using the Mouse Cytokine assay kit according to the manufacturer's specifications (Integrated Biotech Solutions, IBS, Shanghai, China). Levels of IL-2, IL-6, IL-10, MCP-1, IFN-γ, and TNF-α were measured. A total of 1500 events were recorded for each preparation.

**Single cell RNA sequencing.** For scRNA-seq, we followed the protocol described in the Chromium™ Single Cell 3 Reagent Kit (V3 Chemistry) to obtain four single-cell libraries for Illumina sequencing. Peripheral lymphocytes from blood of DHA-treated and CMC control mice were isolated conventionally, centrifuged, and washed with PBS. Next, clean cells were separated using a chromium controller with a 10× Genomics microfluidic system. The cells were used as template in emulsion PCR and single-cell complementary DNA (cDNA) libraries were subsequently constructed and sequenced using an Illumina HiSeq 4000 sequencing

system. Each group comprised two biological replicates; therefore, a total of four cDNA libraries (two each from the DHA-treated group and CMC-treated control group) were constructed.

**Bioinformatic analysis**. The 10× CellRanger pipeline (https://github.com/10XGenomics/cellranger) was used to convert raw data, such as read alignment, bar code counting, and UMI counting. EmptyDrops was then used to filter null cells. Furthermore, Seurat was used to remove low-quality cells (<200 genes/cell, <10 cells/gene, and >10% mitochondrial genes). FindMarkers function was used to identify DEGs for cell subsets between different groups. We followed the general SCENIC workflow from gene filtration to the binarization of transcription factor activity described on the web (https://www.aertslab.org/#scenic).

**Binding of DHA to c-Fos analyzed by biolayer interferometry**. To probe the specific binding of DHA to c-Fos, Interaction experiments were conducted using biolayer interferometry (Octet Red). Recombinant murine proto-oncogene c-Fos was purchased from cusabio (CSB-EP008790MO). His-tagged recombinant proto-oncogene c-Fos as a ligand was diluted in PBS and immobilized at 50 μg/mL on nickel-NTA biosensors for 1200 s. DHA was firstly dissolved in absolute ethanol and then diluted in PBS from the maximum concentration of 6000 μM with a 10-fold gradient dilution for total 3 points. The binding between DHA and c-Fos proteins was analyzed after association for 60 s and then dissociation for 90 s at 30 °C. The data was used in the Octet data analysis software.

**Molecular docking of the binding of DHA to c-Fos**. Molecular docking simulations were performed using the AutoDock MGL Tools (version 1.5.6) program and docking poses generated from Autodock Vina (version 1.1.2) from PDB 2WT7 (Crystal structure of the bZIP heterodimeric complex formed by MafB and c-Fos with DNA) and PDB 1FOS (two human c-Fos-c-Jun-DNA complexes). Docking analyses on the heterodimeric complexes of c-Fos from human and mouse with or without a DNA template was performed. A standard precision docking parameter was set, and 100 ligand poses per docking were run. These structures both single c-Fos complex or bound to DNA were used to dock DHA. For the murine c-Fos complexes with DNA and without DNA, the dimension of the center point coordinate was set $36.044 \times -44.162 \times -1.244$ and $38.109 \times -44.162 \times -1.244$, respectively. For the human c-Fos complexes with DNA and without DNA, the dimension of the center point coordinate was $27.260 \times 5.327 \times -20.249$ and $27.204 \times 6.585 \times -18.664$, respectively. Image produced using PyMol and LigPlot.

**Statistics and reproducibility**. All statistical analyses were performed using the R program (version 4.0.3). The significance of experimental results was assessed using one-way ANOVA (for multiple groups or Student's t-test. All data are presented as mean standard deviation with a $P$ value of 0.05 considered statistically significant. In all cases, replicates are biologically independent samples. Sample size and number of replicates was determined based on previous studies.

**Reporting summary**. Further information on research design is available in the Nature Portfolio Reporting Summary linked to this article.

## Data availability

All data are available in the main text or the supplementary materials. Source data of all plots can be found in Supplementary Data 3. Uncropped Western Blot images are given in Supplementary Fig. 7. The single cell RNA-Seq data analyzed in this study are available on the National Center for Biotechnology Information (NCBI) database under the BioProject PRJNA913669 (SRR22805299 and SRR22805298).

## Code availability

The Seurat was created using the R package available via CRAN-The Comprehensive R Archive Network: https://cloud.r-project.org/web/packages/Seurat/index.html. The source code was from https://github.com/satijalab/seurat/.

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

## Acknowledgements

This research was supported by the National Nature and Science Foundation of China (grant number 82030060), CAMS Innovation Fund for Medical Sciences (CIFMS) (grant number 2019-I2M-5-042).

## Author contributions

Q.L. performed most experiments, analyzed the data, and wrote the first draft of the manuscript. N.J. supervised the study. Z.Y., Z.S., and Q.Y. provided assistance with the animal experiments. Y.L. performed all the molecular docking studies. X.S., Y.F., and R.C. assisted with the flow cytometry experiments and biolayer interferometry. Q.C. conceived the study, analyzed the data, and finalized the manuscript.

## Competing interests

The authors declare no competing interests.
