## [Peer Review File · Communications Biology]

Reviewers' comments:

Reviewer #1 (Remarks to the Author):

This is an interesting work. DHA can promote Treg cell expansion and inhibit B cell expansion has been fully disclosed in the author's previous article (Sci. China Life Sci. 2020, 63, 737-749; Sci. China Life Sci. 2022, DOI: 10.2139/ssrn.3927076). In the work, the authors further found that DHA can regulate Treg and B cell expansion through upregulating C-fos. As the contents of Figure 1 to Figure 4 almost duplicating previous fact that DHA can promote Treg cell expansion and inhibit B cell expansion in vivo and in vitro, and some same experiments have been tested in previous articles (Sci. China Life Sci. 2020, 63, 737-749; Sci. China Life Sci. 2022, DOI: 10.2139/ssrn.3927076), it is inappropriate to expend so much space on what was clearly found. Some data of Figure 1 to Figure 4 is better to be provided in supplementary materials. And, some problems also need to be solved.

1) The authors mentioned that DHA could selectively promote Treg proliferation, meanwhile, suppress B cell expansion. Previous data shown that DHA could affect other Th cells and it is inaccurate to use the word "selectively"; As the same, c-Fos-DHA complex, it just comes from the docking and is not clear that it really exists.

2) The authors claimed that DHA curbed LPS- and plasmodium berghei ANKA-induced inflammation by increasing Treg function. In fact, in Figure 3b, CTLA-4 (CD152) is not a specific or well recognized marker of Treg, the change of CD4+CD152+ can not indicate the change of Treg, and other indicators such as TNF-alpha, IFN-gamma, MCP-1, IL-6 and IL-2 are nonspecific markers, which also can not well support that this effect comes from tregs. So the changes in some specific markers of Treg should be further provided. In addition, please check the name of groups are right? Figure 3a and 3b, without the data of ANKA-treated groups.

3) In section of "DHA upregulated C-fos expression in plasma cells and Tregs cells", the effect of DHA on CXCR5+CD25+ T follicular cells was induced and it not very relevant to the topic.

4) The single cell analysis indicated DHA upregulated c-fos expression in B cells and treg cells, however, no experimental data clearly verified that c-fos expression was really upregulated in B cells and treg cells, c-fos expression in purified B cells and treg cells should be provided.

5) In Figure 6, all indicators were tested in splenic cells and these data can not clearly support the conclusion, since splenic cells is a mixed system and the contribution to this effect does not exclude from other cells such as NK cells, CD8 T cells.

Reviewer #2 (Remarks to the Author):

Known in the field based on previous literatures:

1. Dihydroartemisinin (DHA) is an active metabolite of artemisinin. DHA, the clinically relevant artemisinin kills malaria parasites via two-way, protein damage and compromising parasite proteasome function. DHA also reported in anti-leishmaniasis.

2. DHA has exhibited anticancer effects on many types of tumors, including lung, breast, prostate, ovarian, digestive system tumors.

3. In general, DHA has been established to have anticancer effects, including inhibiting proliferation, inducing apoptosis, promoting immune function, inducing autophagy and endoplasmic reticulum stress.

In this manuscript authors reported following findings:

I have gone through the manuscript titled "Dihydroartemisinin (DHA) imposes positive and negative regulation on Treg and B cells via direct interaction and activation of c-Fos transcription factor". Manuscript describes the bilateral immunoregulatory mechanism of DHA and their application in the autoimmune disease. To investigate the function of the DHA, the authors have shown that DHA inhibit the inflammation induced by both lipopolysaccharide (LPS) and Plasmodium berghei. Authors are performed and reported following findings-

1. DHA suppressed B cell proliferation and enhanced Treg expansion and their associated genes

2. DHA inhibit the inflammation induced by both LPS and Plasmodium berghei ANKA via activation

of Treg cells.

3. DHA interact with c-FOS and induced c-Fos expression in both Treg and B cells.

The data presented are interesting and generally supportive of the conclusions drawn. There are, however, several issues that require the authors' attention. The following minor suggestions if incorporated could help in the better understanding of the significance of the work and implications.

Minor Concerns:

1. Authors have reported DHA treatment significantly decreased parasitemia. What was the parasitemia in ANKA+DHA vs ANKA? Are parasites were cleared from DHA treated mice? Please include the parasitemia graph in supplementary figure if you have.
2. How did you choose to take splenic tissues and blood assessment on day 26? Since, in various previous studies authors measured the inflammatory cytokines at day 4-6 in Plasmodium yoelii infection. How and why did you decide to measure inflammatory cytokines at day 26?
3. What was the effect of parasites on spleen? Have you seen spleen enlargement in parasites treated mice and further reduce in DHA cotreated mice?
4. The genetic background of both host and parasites can greatly influence malaria disease severity, and almost identical genome led to different disease phenotypes. For examples, BALB/c mice infected with the parasites Plasmodium yoelii yoelii 17XL (YM) die within 7 days of post infection, whereas isogenic strain P. y. yoelii 17XNL recover from infection. What was the reason to use the Plasmodium berghei ANKA strain?
5. The best finding is DHA bound to c-Fos on the DNA binding domain. Explain how your study is different from rest excluding this result.
6. The sentence and line 379 seem incomplete. The sentence is- Here, we reported that DHA selectively increased the proportion of Treg while decreased the number of plasma cells both in healthy (Fig. 1 and 2). Please mention another group.

Reviewer #3 (Remarks to the Author):

The authors reported that the bilateral effect anti-malaria drug DHA on B cells and Treg cells, and elucidated the underlying mechanism of DHA to induce B cell apoptosis and Treg expansion, which was contributed to the interaction of DHA with specific transcription factor c-Fos. Furthermore, its negative effect on B cells and positive effect on Treg were verified in two immunomodulating models. It sounds interesting, and explains why DHA could be used to treat autoimmune diseases. However, I still have several comments, which are needed to be addressed by authors before the manuscript could be published.

Major

1. It is puzzle to me that DHA has different effect on Treg and B cells through acting on the same transcription factor c-Fos. As author's results mainly came from the in vivo treatment of mice with DHA, and no in vitro effects of DHA on B cells have been performed. Therefore, the indirect effect of DHA on B cells could not be excluded. In addition, Treg could modulate the antibody responses, and authors also found the expansion of CXCR5+ CD25+ T follicular regulatory cells in their SRBC immunization experiments. Is it possible for DHA to indirectly suppress B cell response through promoting Treg cell expansion? This should be explained or addressed in the discussion section.
2. For testing the modulation effect of DHA on Treg, LPS-induced sepsis model is enough. I do not think the P.berghei infection model is appropriate to test the positive effect of DHA on Treg, as the killing effect of DHA on P.berghei could not be excluded. Therefore, I suggest to remove the data of P.berghei infection model.
3. As we know, DHA is the frontline antimalarial drug. However, in the present study, authors claimed that DHA could also suppress host immune responses, which would counteract the treat effect of DHA on malaria patients. Is there any suggestion for authors to optimize the treatment effect of DHA on malaria patients?

Minor:

1. Could anti-CD4 and Ki67 label specifically identify the white pulp and GC area?

2. Supplementary Figure 5 a and b, which demonstrated the direct interaction of DHA and c-Fos, are important data to support the conclusion of the present study. I strongly suggest to include these data in the main text, but not the supplemental materials.
3. Line 90. The full text of "CMC" should be given for reader to understand.
4. Line 262. Both of Foxp1 and Foxo1 are not found in the heatmap.
5. Line 284. A reference is required.
6. Line 458. A protocol number is needed.

Answer to Questions and Comments of the Reviewers

Reviewer #1:

Comment 1: *This is an interesting work. DHA can promote Treg cell expansion and inhibit B cell expansion has been fully disclosed in the author's previous article (Sci. China Life Sci. 2020, 63, 737-749; Sci. China Life Sci. 2022, DOI: 10.2139/ssrn.3927076). In the work, the authors further found that DHA can regulate Treg and B cell expansion through upregulating C-fos. As the contents of Figure 1 to Figure 4 almost duplicating previous fact that DHA can promote Treg cell expansion and inhibit B cell expansion in vivo and in vitro, and some same experiments have been tested in previous articles (Sci. China Life Sci. 2020, 63, 737-749; 2022, DOI: 10.2139/ssrn.3927076), it is inappropriate to expend so much space on what was clearly found. Some data of Figure 1 to Figure 4 is better to be provided in supplementary materials. And, some problems also need to be solved.*

Answer to Comment 1: We are sorry for the unclearness. In our studies, the immunomodulatory activities of DHA was preliminarily focused on four simple cell populations, including CD4⁺ or CD8⁺ T cells, CD19⁺ B cells and CD49b⁺ NK cells (Sci. China Life Sci. 2020, 63, 737-749). The present work is a continuous and in-depth study on the regulatory mechanism. Thus, the data presented in the manuscript are significant different from that reported in the earlier papers.

In this study, we mainly focused on effect of DHA on immune cells in the blood, while only splenic cells were investigated in the earlier papers. Further, we also provided several novel data and observations with single cell RNA sequencing to

determine the effect of DHA on circulating immune cells in this study. Furthermore, we used new mouse models to determine the effect of DHA on both circulating plasma cells and T_{regs}, as opposed to simply examining splenic plasma cells and T_{regs} in normal mice as in an earlier study.

In fact, Figure 1 showed the effect of DHA on immune cells in the peripheral blood, including the absolute counts of multiple immune cells and the ratio of B cell subgroups. To explain the phenomenon generated in Figure 1, we used a classical immunological model to investigate the inhibitory effect of DHA on antigen-presenting cells-dependent humoral immune responses. Thus, the data in Figure 1-4 represent advanced analysis, which are different from that in the previous articles.

Comment 2: *The authors mentioned that DHA could selectively promote Treg proliferation, meanwhile, suppress B cell expansion. Previous data shown that DHA could affect other Th cells and it is inaccurate to use the word “selectively”; As the same, c-Fos-DHA complex, it just comes from the docking and is not clear that it really exists.*

Answer to Comment 2: We appreciate the constructive comments. As suggested, we have removed the word “selectively” in Abstract and Discussion.

As for the c-FOS-DHA interaction, we explored two experimental approaches to validate the result of molecular docking in **Figure 6 and 7**. Firstly, the c-FOS specific inhibitor (T5224) indeed inhibited the activity of DHA on T_{reg} and B cells (Figure 6). Secondly, the biolayer interferometry assay shown that DHA exhibited high binding

affinity to murine c-Fos, but it did not bind to an irrelevant his-tagged control protein (Figure 7). All these data indicated DHA could directly interact with c-FOS.

Comment 3: *The authors claimed that DHA curbed LPS- and plasmodium berghei ANKA-induced inflammation by increasing Treg function. In fact, in Figure 3b, CTLA-4 (CD152) is not a specific or well recognized marker of Treg, the change of CD4⁺CD152⁺ can not indicate the change of Treg, and other indicators such as TNF-alpha, IFN-gamma, MCP-1, IL-6 and IL-2 are nonspecific markers, which also can not well support that this effect comes from tregs. So the changes in some specific markers of Treg should be further provided. In addition, please check the name of groups are right? Figure 3a and 3b, without the data of ANKA-treated groups.*

Answer to Comment 3: We apologize for the unclearness and misunderstanding. In fact, the ordinate of **Figure 3b** represents the proportion of total CD152⁺ T_{reg} gated off the total CD4⁺ populations. Based on the single-cell sequencing results, we found the expression of CD152 in T_{reg} was overexpressed under DHA treatment (**Supplementary Table S1**), thus we used an additional T_{reg} suppression marker CD152¹⁻⁵ to label T_{reg}. We have revised the **Figure legend** to describe the proportion of total CD152⁺ T_{reg} gated off the total CD4⁺ populations.

We did not used TNF-alpha, IFN-gamma, MCP-1, IL-6 and IL-2 as markers of T_{reg} in this study. Based on the literatures, as far as we are aware, the cytokine responses such as TNF- α and IL-10 only represent one aspect of inflammatory responses^{6,7}. Here, the results here illustrate the shift in cytokine concentrations under DHA treatment in LPS-treated or ANKA-infected mice.

References:

1. Liu M, Kuo F, Capistrano KJ, Kang D, Nixon BG, Shi W, Chou C, Do MH, Stamatiades EG, Gao S, Li S, Chen Y, Hsieh JJ, Hakimi AA, Taniuchi I, Chan TA, Li MO. TGF- β suppresses type 2 immunity to cancer. *Nature*. 2020 Nov;587(7832):115-120. doi: 10.1038/s41586-020-2836-1. Epub 2020 Oct 21.
2. Walker LS. Treg and CTLA-4: two intertwining pathways to immune tolerance. *J Autoimmun*. 2013 Sep;45(100):49-57. doi: 10.1016/j.jaut.2013.06.006. Epub 2013 Jul 10.
3. Zappasodi, R., Serganova, I., Cohen, I.J. et al. CTLA-4 blockade drives loss of Treg stability in glycolysis-low tumours. *Nature* 591, 652–658 (2021). <https://doi.org/10.1038/s41586-021-03326-4>
4. Xuguang Tai, François Van Laethem, Leonid Pobeziński, Terry Guinter, Susan O. Sharrow, Anthony Adams, Larry Granger, Michael Kruhlak, Tullia Lindsten, Craig B. Thompson, Lionel Feigenbaum, Alfred Singer; Basis of CTLA-4 function in regulatory and conventional CD4⁺ T cells. *Blood* 2012; 119 (22): 5155–5163.
5. Emily M. Schmidt, Chun Jing Wang, Gemma A. Ryan, Louise E. Clough, Omar S. Qureshi, Margaret Goodall, Abul K. Abbas, Arlene H. Sharpe, David M. Sansom, Lucy S. K. Walker *The Journal of Immunology* January 1, 2009, 182 (1) 274-282
6. Tan H., Zhao J., Zhang H., Zhai Q., Chen W. Novel strains of *Bacteroides fragilis* and *Bacteroides ovatus* alleviate the LPS-induced inflammation in mice. *Appl. Microbiol. Biotechnol.* 2019;103:2353–2365.
7. Chung L., Orberg E.T., Geis A.L., Chan J.L., Fu K., Shields C.E., Dejea C.M., Fathi P., Chen J., Finard B.B., et al. *Bacteroides fragilis* Toxin Coordinates a Pro-carcinogenic Inflammatory Cascade via Targeting of Colonic Epithelial Cells. *Cell Host Microbe*. 2018;23:203–214.

Comment 4: *In section of “DHA upregulated C-fos expression in plasma cells and Tregs cells”, the effect of DHA on CXCR5+CD25+ T follicular cells was induced and it not very relevant to the topic.*

Answer to Comment 4: We agree that this section is not critical for the manuscript and have removed it as suggested. **Figure 5** and this section have been revised.

Comment 5: *The single cell analysis indicated DHA upregulated c-fos expression in B cells and treg cells, however, no experimental data clearly verified that c-fos expression was really upregulated in B cells and treg cells, c-fos expression in purified B cells and treg cells should be provided.*

Answer to Comment 5: We appreciate the constructive comments. Because of the rarity of these cells in the peripheral blood, the amount of c-Fos expressed in Treg and circulating plasma cells was estimated by flow cytometry referred to a previously reported method¹⁻³. These results were in line with that from the single cell sequencing, suggesting that c-Fos expression was really upregulated in PC B cells and T_{regs} by DHA. Figure 6 has been revised with now results been presented in Fig. 6 b and c. More text has been added to the Result and Method.

References:

1. Clark, Carolyn E., Milena Hasan, and Philippe Bousso. "A role for the immediate early gene product c-fos in imprinting T cells with short-term memory for signal summation." *PLoS One* 6.4 (2011): e18916.
2. Bendfeldt, Hanna, et al. "Stable IL-2 decision making by endogenous c-Fos amounts in peripheral memory T-helper cells." *Journal of Biological Chemistry* 287.22 (2012): 18386-18397.
3. Hop, Huynh T., et al. "The key role of c-Fos for immune regulation and bacterial dissemination in Brucella infected macrophage." *Frontiers in Cellular and Infection Microbiology* 8 (2018): 287.

Comment 6: In Figure 6, all indicators were tested in splenic cells and these data cannot clearly support the conclusion, since splenic cells is a mixed system and the contribution to this effect does not exclude from other cells such as NK cells, CD8 T cells.

Answer to Comment 6: We are sorry for the misunderstanding. Actually, the data in Figure 6 represent c-Fos expression in both T_{reg} and B cells from peripheral blood, not from the spleen. One of the conclusions of this work is that DHA can upregulate c-Fos expression and enhance its interaction with target genes in both Treg and B cells with bilateral cell fates. Both the spleen and peripheral blood are highly complicated niches

for cross-talk of multifarious immune cells. Given the prominent role of cross-talk in multifarious immune cells, it is important to consider that it is difficult to discriminate whether the positive and negative regulation on T_{reg} and B cells are activated directly and/or indirectly by DHA. However, the data from the experiment with c-FOS inhibitor T5224 do indicates that the immunoregulatory activity of DHA in peripheral blood only in the presence of c-FOS (**Figure 6 g and h**).

Furthermore, in **Figure 1**, we have provided the changes in the ratio of NK cells and CD8 T cells. The absolute counts of peripheral blood CD8⁺ T cells and NK cells were reduced in the DHA-treated group compared to those in the control group (**Figure 1 c and d**). More text has been added to the Discussion.

Reviewer #2:

Comment 1: *The data presented are interesting and generally supportive of the conclusions drawn. There are, however, several issues that require the authors' attention. The following minor suggestions if incorporated could help in the better understanding of the significance of the work and implications.*

Answer to Comment 1: We appreciate very much the positive comments.

Comment 2: *Authors have reported DHA treatment significantly decreased parasitemia. What was the parasitemia in ANKA+DHA vs ANKA? Are parasites were cleared from DHA treated mice? Please include the parasitemia graph in supplementary figure if you have.*

Answer to Comment 2: We are sorry for the misunderstanding. In our previous findings, we found that pre-treatment with DHA could enhance host innate immunity against parasite infection. Thus, we orally administered DHA 26 days **BEFORE** parasite infection. We found the pretreatment with DHA indeed prolonged mouse survival time. In this study, our aim was to dissect the immune-modulatory mechanism of DHA. We found that DHA prolonged the survival time of LPS and ANKA co-infected mice. This may be closely related to the immunomodulatory function of DHA. The result section and **Figure 3c** legend has been revised. A new **Supplementary Figure 2b** has been added in Supplementary Information

Comment 3: *How did you choose to take splenic tissues and blood assessment on day 26? Since, in various previous studies authors measured the inflammatory cytokines at day 4-6 in Plasmodium yoelii infection. How and why did you decide to measure inflammatory cytokines at day 26?*

Answer to Comment 3: We are sorry for the unclearness. Timing of splenic tissues and blood assessment was chosen on the basis of findings from our previous experiments (*Sci. China Life Sci.* 2020, 63, 737-749). We did not perform cytokine experiments at day 26. According to our previous observation, mice showed the most featured cytokine responses after 7 days infection by *P. berghei*. The cytokine responses were investigated at day 7, but not at day 26. Additionally, in various previous studies authors measured the inflammatory cytokines at day 6-8 in *P. berghei* ANKA infection and LPS infection¹⁻⁴. Therefore, we chose day 7 for the

measure of inflammatory cytokines. The legend of **Figure 3 has been revised.**

References:

1. Syarifah Hanum P., Masashi Hayano, Somei Kojima, Cytokine and chemokine responses in a cerebral malaria - susceptible or - resistant strain of mice to Plasmodium berghei ANKA infection: early chemokine expression in the brain, *International Immunology*, Volume 15, Issue 5, May 2003, Pages 633–640.
2. Settles EW, Moser LA, Harris TH, Knoll LJ. Toxoplasma gondii upregulates interleukin-12 to prevent Plasmodium berghei-induced experimental cerebral malaria. *Infection and Immunity*. 2014 Mar;82(3):1343-1353.
3. Dickmann, L. J., McBride, H. J., Patel, S. K., Miner, K., Wienkers, L. C., & Slatter, J. G. (2012). Murine collagen antibody induced arthritis (CAIA) and primary mouse hepatocyte culture as models to study cytochrome P450 suppression. *Biochemical pharmacology*, 83(12), 1682–1689.
4. Suh HN, Kim YK, Lee JY, Kang GH, Hwang JH. Dissect the immunity using cytokine profiling and NF- κ B target gene analysis in systemic inflammatory minipig model. *PLoS One*. 2021 Jun 4;16(6):e0252947.

Comment 4: *What was the effect of parasites on spleen? Have you seen spleen*

enlargement in parasites treated mice and further reduce in DHA cotreated mice?

Answer to Comment 4: Firstly, we indeed observed enlargement of the spleen in *P. berghei* ANKA infected mice¹. Further, no enlargement was observed in mice treated with DHA early in the infection by *P. berghei* ANKA, which was likely attributed to the therapeutic effect of DHA to the parasites. However, with the prolonged DHA treatment in the later stages of infection, enlarged spleens were observed, which was likely due to the cell population expansion induced by DHA.

Reference:

1. Zhang, Y., Jiang, N., Zhang, T. et al. Tim-3 signaling blockade with α -lactose induces compensatory TIGIT expression in Plasmodium berghei ANKA-infected mice. *Parasites Vectors* 12, 534 (2019).

Comment 5: *The genetic background of both host and parasites can greatly influence*

malaria disease severity, and almost identical genome led to different disease phenotypes. For examples, BALB/c mice infected with the parasites Plasmodium yoelii yoelii 17XL (YM) die within 7 days of post infection, whereas isogenic strain P. y. yoelii 17XNL recover from infection. What was the reason to use the Plasmodium berghei ANKA strain?

Answer to Comment 5: We appreciate and completely agree with the comment. *P. berghei* ANKA strain has been commonly used to dissect malaria pathogenesis, including cerebral malaria, one of the most severe clinical forms of *P. falciparum* infection. The purpose of this study was to investigate the immunoregulatory effect of DHA, we only chose *P. berghei* ANKA strain.

Comment 6: *The best finding is DHA bound to c-Fos on the DNA binding domain. Explain how your study is different from rest excluding this result.*

Answer to Comment 6: We appreciate the comment. One of the conclusions of this work is that the upregulation of c-Fos expression by DHA and enhancement of its interaction with target DNA binding domain in both Treg and B cells with bilateral cell fates. To the best of our knowledge, this study is the first to show the association between DHA and c-Fos on the DNA binding domain. More text has been added to the Discussion.

Comment 7: *The sentence and line 379 seem incomplete. The sentence is- Here, we reported that DHA selectively increased the proportion of Treg while decreased the*

number of plasma cells both in healthy (Fig. 1 and 2). Please mention another group.

Answer to Comment 7: We appreciate the constructive comments. We have added another group in this sentence.

Reviewer #3:

Comment 1: *The authors reported that the bilateral effect anti-malaria drug DHA on B cells and Treg cells, and elucidated the underlying mechanism of DHA to induce B cell apoptosis and Treg expansion, which was contributed to the interaction of DHA with specific transcription factor c-Fos. Furthermore, its negative effect on B cells and positive effect on Treg were verified in two immunomodulating models. It sounds interesting, and explains why DHA could be used to treat autoimmune diseases. However, I still have several comments, which are needed to be addressed by authors before the manuscript could be published.*

Answer to Comment 1: We appreciate the positive comments.

Comment 2: *It is puzzle to me that DHA has different effect on Treg and B cells through acting on the same transcription factor c-Fos. As author's results mainly came from the in vivo treatment of mice with DHA, and no in vitro effects of DHA on B cells have been performed. Therefore, the indirect effect of DHA on B cells could not be excluded. In addition, Treg could modulate the antibody responses, and authors also found the expansion of CXCR5+ CD25+ T follicular regulatory cells in their SRBC immunization experiments. Is it possible for DHA to indirectly suppress B cell response through promoting Treg cell expansion? This should be explained or addressed in the discussion section.*

Answer to Comment 2: We appreciate the question. We have removed this result also as suggested by reviewer 1. One of the key conclusions of this work is that the upregulation of c-Fos expression by DHA and enhancement of its interaction with

target genes in both T_{reg} and PC B cells with bilateral cell fates. Both the spleen and peripheral blood are highly complicated niches for cross-talk of multifarious immune cells. Given the prominent role of cross-talk in multifarious immune cells, it is important to consider that it is difficult to discriminate whether the positive and negative regulation on Treg and PC B cells are activated directly and/or indirectly by DHA. However, based upon the c-FOS inhibitor T5224 outcomes, we could affirm that the immunoregulatory functions of DHA in peripheral blood only initiated under presence c-FOS conditions (**Figure 6 g and h**). More text has been added to the discussion.

Comment 3: *For testing the modulation effect of DHA on Treg, LPS-induced sepsis model is enough. I do not think the P.berghei infection model is appropriate to test the positive effect of DHA on Treg, as the killing effect of DHA on P.berghei could not be excluded. Therefore, I suggest to remove the data of P.berghei infection model.*

Answer to Comment 3: We appreciate this comment. Since the main function of DHA is an antimalarial drug. Thus, we wanted to investigate whether DHA could both kill the parasite and reduce the inflammatory response in the LPS combined with *P. berghei* ANKA infection model. As shown in our **Figure 3d**, mice infected with *P. berghei* ANKA or injected with LPS both caused elevated TNF- α . DHA treatment reduced the level of TNF- α in the serum of both LPS-treated, ANKA-treated mice or LPS combined with ANKA- treated mice. This suggests that while LPS accelerates ANKA death, DHA is simultaneously effective in reducing comorbidity. We are very

sorry the unclearness and the section title and content have been revised.

Comment 4: *As we know, DHA is the frontline antimalarial drug. However, in the present study, authors claimed that DHA could also suppress host immune responses, which would counteract the treat effect of DHA on malaria patients. Is there any suggestion for authors to optimize the treatment effect of DHA on malaria patients?*

Answer to Comment 4: We appreciate this question. The artemisinin-based combination therapy dihydroartemisinin-piperaquine (DP) is the leading candidate for intermittent preventive treatment (IPT) in young children in malaria-endemic regions with widespread sulfadoxine-pyrimethamine (SP) resistance. Large studies have not identified significant toxicities associated with IPT with DP in children, even though plasma piperaquine concentration has been positively associated with lengthening of the corrected QT interval ^{1,2}. Our study indicated that it is more important to monitor changes in immune indicators in malaria patients during prolonged treatment with DHA. Corresponding **DISCUSSIONS** have been added to the manuscript.

References:

1. Chan, X. H. S. et al. Risk of sudden unexplained death after use of dihydroartemisinin-piperaquine for malaria: a systematic review and Bayesian meta-analysis. *Lancet Infect. Dis.* 18, 913–923 (2018).
2. Chotsiri, P. et al. Population pharmacokinetics and electrocardiographic effects of dihydroartemisinin-piperaquine in healthy volunteers. *Br. J. Clin. Pharm.* 83, 2752–2766 (2017).

Comment 5: *Could anti-CD4 and Ki67 label specifically identify the white pulp and GC area?*

Answer to Comment 5: We appreciate the constructive comments. CD4 and Ki67 have been widely used to show white pulp and GC area in mice^{1,2}.

References:

1. Wang, Y., & Carter, R. H. (2005). CD19 regulates B cell maturation, proliferation, and positive selection in the FDC zone of murine splenic germinal centers. *Immunity*, 22(6), 749–761. <https://doi.org/10.1016/j.immuni.2005.04.012>
2. Rahman, Z. S., Rao, S. P., Kalled, S. L., & Manser, T. (2003). Normal induction but attenuated progression of germinal center responses in BAFF and BAFF-R signaling-deficient mice. *The Journal of experimental medicine*, 198(8), 1157–1169.

Comment 6: *Supplementary Figure 5 a and b, which demonstrated the direct interaction of DHA and c-Fos, are important data to support the conclusion of the present study. I strongly suggest to include these data in the main text, but not the supplemental materials.*

Answer to Comment 6: We appreciate the constructive comments. We have moved these data in the main text. A new Figure 7 has been added.

Comment 7: *Line 90. The full text of “CMC” should be given for reader to understand.*

Answer to Comment 7: We appreciate the constructive comments. The full text of “CMC” has been added.

Comment 8: *Line 262. Both of Foxp1 and Foxo1 are not found in the heatmap.*

Answer to Comment 8: Foxp1 and Foxo1 are not TOP10 transcription factors, so they are not shown in the heatmap. We have removed this sentence

Comment 9: *Line 284. A reference is required.*

Answer to Comment 9: We appreciate the constructive comments. We have removed this result as suggested by reviewer1.

Comment 10: *Line 458. A protocol number is needed.*

Answer to Comment 10: We removed the Ethical statement due to double blind review. We have added protocol number in Method.

Reviewers' comments:

Reviewer #1 (Remarks to the Author):

Although the article has been revised, there is still a lack of direct evidence that DHA can upregulate C-FOS in B cells and Treg cells to inhibit B cell proliferation and promote Treg expansion. Validation of the effects of DHA in vitro with purified B cells and Treg cells, such as those derived from the spleen or lymph nodes, is essential and can be done successfully in any laboratory. The authors tried to demonstrate this effect by single-cell sequencing analysis and flow MFI results, but these results were from complex systems and were not convincing. Also, other problems:

1) The author intends to use the LPS-induced inflammation model and the LPS + ANKA infection inflammation model to evaluate the effect of DHA, but Figure 3d-3i showed the deficiency of LPS + ANKA model group.

2) Figure 5f, co-staining of Ki67 and c-Fos on spleen sections showed that Ki67 and c-Fos were localized to different cells and could not support the changes in c-FOS on B cells.

3) No information was provided for the gating strategies of MFI of c-fos in plasma cells and MFI of c-fos in Treg.

Reviewer #2 (Remarks to the Author):

I have gone through the revised manuscript titled " Dihydroartemisinin imposes positive and negative regulation on Treg and B cells via direct interaction and activation of c-Fos transcription factor". Manuscript describes the bilateral immunoregulatory mechanism of DHA and their application in the autoimmune disease. Authors have included or cleared the raised question by me, and the data presented are interesting and generally supportive of the conclusions drawn. The manuscript could be of interest to the field of immunoregulatory mechanism of DHA and its application in the treatment of autoimmune diseases.

Reviewer #3 (Remarks to the Author):

The authors have answered all of my questions, and the manuscript has been significantly improved.

As LPS-induced sepsis model is enough to demonstrate the modulation effect of DHA on Treg, I suggest to remove the data of P.berghei infection model into supplemental materials.

Answer to Questions and Comments of the Reviewers

Reviewer #1:

Comment 1: *Although the article has been revised, there is still a lack of direct evidence that DHA can upregulate C-FOS in B cells and Treg cells to inhibit B cell proliferation and promote Treg expansion. Validation of the effects of DHA in vitro with purified B cells and Treg cells, such as those derived from the spleen or lymph nodes, is essential and can be done successfully in any laboratory. The authors tried to demonstrate this effect by single-cell sequencing analysis and flow MFI results, but these results were from complex systems and were not convincing.*

Answer to Comment 1: We are sorry for the confusion. In the present study, the immunomodulatory effect of DHA was primarily on Treg and circulating plasma cells, not in B cells. We have replaced the word “B cells” in the text. The title of the manuscript has also been revised. Further, as we know and others report, B cells cultured alone were unable to proliferate and differentiate into plasma cells, and even underwent apoptosis ($81\% \pm 3.5\%$)¹. Several papers reported that B cell can proliferate and differentiate into plasma cells under CD40L, IL-4, BAFF and IL-21 co-stimulated conditions². But this experiment needs two special cell line CD40L/Baff-expressing BALB/c 3T3 fibroblasts (40LB)-IL4 and 40LB-IL21 cell lines. These cell lines were stored in the

RIKEN BioResource Research Center (Japan, Cell number: RCB5304). But we unable to get the cells from Japan at this stage. Therefore, we cannot verify the effects of DHA *in vitro* with purified circulating plasma cells.

Many studies have demonstrated that DHA alone or in combination with prednisone significantly inhibited Th17 cell differentiation while induced Treg cell differentiation *in vitro*⁴. In additional, DHA has been reported that it greatly promoted Treg cell generation *in vitro*⁵. Therefore, DHA act directly on Treg cell. But the precise molecular mechanisms by which DHA does this *in vivo* has not been well defined. In this article, we proposed the molecular mechanisms of immunomodulatory effect of DHA *in vivo*.

Reference:

1. Wei, Y., Lao, X. M., Xiao, X., Wang, X. Y., Wu, Z. J., Zeng, Q. H., Wu, C. Y., Wu, R. Q., Chen, Z. X., Zheng, L., Li, B., & Kuang, D. M. (2019). Plasma Cell Polarization to the Immunoglobulin G Phenotype in Hepatocellular Carcinomas Involves Epigenetic Alterations and Promotes Hepatoma Progression in Mice. *Gastroenterology*, 156(6), 1890–1904.e16.
2. Haniuda K, Kitamura D. Induced Germinal Center B Cell Culture System. *Bio Protoc.* 2019 Feb 20;9(4):e3163. doi: 10.21769/BioProtoc.3163. PMID: 33654969; PMCID: PMC7854248.

3. Nojima, T., Haniuda, K., Moutai, T. et al. In-vitro derived germinal centre B cells differentially generate memory B or plasma cells in vivo. *Nat Commun* 2, 465 (2011).
4. Chen, Y, Tao, T, Wang, W, Yang, B, Cha, X. Dihydroartemisinin attenuated the symptoms of mice model of systemic lupus erythematosus by restoring the Treg/Th17 balance. *Clin Exp Pharmacol Physiol*. 2021; 48: 626– 633.
5. Zhao YG, Wang Y, Guo Z, Gu AD, Dan HC, Baldwin AS, Hao W, Wan YY. Dihydroartemisinin ameliorates inflammatory disease by its reciprocal effects on Th and regulatory T cell function via modulating the mammalian target of rapamycin pathway. *J Immunol*. 2012 Nov 1;189(9):4417-25. doi: 10.4049/jimmunol.1200919. Epub 2012 Sep 19. PMID: 22993204; PMCID: PMC3478428.

Comment 2: *The author intends to use the LPS-induced inflammation model and the LPS + ANKA infection inflammation model to evaluate the effect of DHA, but Figure 3d-3i showed the deficiency of LPS + ANKA model group.*

Answer to Comment 2: We appreciate the constructive comments. The sera from all groups were collected at day 7 after DHA treatment and the levels of the cytokines were determined by cytometric bead array (CBA).

However, LPS induces premature death in mice infected with *P. berghei* ANKA (Figure 3c). Therefore, Figure 3d-3i showed the deficiency of LPS + ANKA model group.

Comment 3: *Figure 5f, co-staining of Ki67 and c-Fos on spleen sections showed that Ki67 and c-Fos were localized to different cells and could not support the changes in c-FOS on B cells.*

Answer to Comment 3: We appreciate the constructive comments. Mature secondary germinal centers were identified based on the size and segregation in the dark and light zones, easily recognizable based on the Ki67 staining pattern. In the solvent control groups, mature secondary germinal centers expressed only trace amount of c-Fos. However, in the DHA-treated group, the germinal centers are atrophic, Ki67 was very much less expressed, but c-Fos was highly expressed on the contrary. That was the reason that Ki67 was not clearly observed the DHA-treated B cells. In additions, it has been reported that, in c-Fos overexpressed mice, the expression of Ki67 was overwhelmed by c-Fos in a mutually exclusive manner.

Reference:

1. Theil, D., Weckbecker, G. (2019). Imaging Mass Cytometry and Single-Cell Genomics Reveal Differential Depletion and Repletion of B-Cell Populations Following Ofatumumab Treatment in Cynomolgus

Monkeys. *Frontiers in immunology*, 10, 1340.

2. Inada, K., Okada, S., Phuchareon, J., Hatano, M., Sugimoto, T., Moriya, H., & Tokuhisa, T. (1998). c-Fos induces apoptosis in germinal center B cells. *Journal of immunology (Baltimore, Md.: 1950)*, 161(8), 3853–3861.

Comment 4: *No information was provided for the gating strategies of MFI of c-fos in plasma cells and MFI of c-fos in Treg.*

Answer to Comment 4: We appreciate the constructive comments. We have added the gating strategies in Method sections.

Reviewer #2:

Comment 1: *I have gone through the revised manuscript titled "Dihydroartemisinin imposes positive and negative regulation on Treg and B cells via direct interaction and activation of c-Fos transcription factor". Manuscript describes the bilateral immunoregulatory mechanism of DHA and their application in the autoimmune disease. Authors have included or cleared the raised question by me, and the data presented are interesting and generally supportive of the conclusions drawn. The manuscript could be of interest to the field of immunoregulatory mechanism of DHA and its application in the treatment of autoimmune diseases.*

Answer to Comment 1: We appreciate very much the positive comments.

Reviewer #3:

Comment 1: *The authors have answered all of my questions, and the manuscript has been significantly improved. As LPS-induced sepsis model is enough to demonstrate the modulation effect of DHA on Treg, I suggest to remove the data of P.berghei infection model into supplemental materials.*

Answer to Comment 1: We appreciate the positive comments. We have moved the data of *P. berghei* infection model into supplemental section. A new Supplementary Figure 2 c-h has been added in the Supplementary information.

REVIEWERS' COMMENTS:

Reviewer #1 (Remarks to the Author):

The authors have answered all of my questions, and the manuscript has been well improved.